# Risk-informed representative earthquake scenarios for Valparaíso and Viña del Mar, Chile

Hugo Rosero-Velásquez[1], Mauricio Monsalve[2,3], Juan Camilo Gómez Zapata[4,5], Elisa Ferrario[2,3,6], Alan Poulos[7], Juan Carlos de la Llera[3,8], and Daniel Straub[1]

[1]Engineering Risk Analysis Group, TU Munich, Arcisstr. 21, 80333 Munich, Germany
[2]School of Engineering, Pontificia Universidad Católica de Chile, Santiago, Chile
[3]Research Center for Integrated Disaster Risk Management (CIGIDEN), ANID/FONDAP/1522A0005, Santiago, Chile
[4]Seismic Hazard and Risk Dynamics, GFZ German Research Centre for Geosciences, 14473 Potsdam, Germany
[5]Institute for Geosciences, University of Potsdam, Karl-Liebknecht-Str. 24–25, 14476 Potsdam, Germany
[6]Ricerca sul Sistema Energetico - RSE S.p.A., Milano, Italy
[7]Department of Civil and Environmental Engineering, Stanford University, Stanford, California, U.S.A
[8]Department of Structural and Geotechnical Engineering, Pontificia Universidad Católica de Chile, Santiago, Chile

**Correspondence:** Hugo Rosero-Velásquez (hugo.rosero@tum.de)

**Abstract.** Different risk management activities, such as land-use planning, preparedness and emergency response, utilize scenarios of earthquake events. A systematic selection of such scenarios should aim at finding those that are representative of a certain severity, which can be measured by its consequences to the exposed assets. For this reason, it has been proposed to define a representative scenario as the most likely one leading to a loss with a specific return period, e.g., the 100-year loss. We adopt this definition and develop enhanced algorithms for determining such scenarios for multiple return periods. With this approach, we identify representative earthquake scenarios for the return periods of 50 yr, 100 yr, 500 yr and 1000 yr in the Chilean communes of Valparaíso and Viña del Mar, based on a synthetic earthquake catalog of $20\,000$ scenarios on the subduction zone with magnitude $M_w \geq 5.0$. We consider separately the residential building stock and the electrical power network, and identify and compare earthquake scenarios that are representative for these systems. Because the representative earthquake scenarios are defined in terms of the annual loss exceedance rates, they vary in function of the exposed system. The identified representative scenarios for the building stock have epicenters located not further than 30 km from the two communes and magnitudes ranging between 6.0 and 7.0. The epicenter locations of the earthquake scenarios representative for the electrical power network are more spread out, but not further than 100 km away from the two communes, and with magnitudes ranging between 7.0 and 9.0. For risk management activities, we recommend considering the identified scenarios together with historical events.

## 1 Introduction

Due to the complexity of earthquake events and the response of infrastructure and society to these events, risk managers analyze potential impacts of strong seismic events and test risk management capacities through representative earthquake scenarios (e.g., Salgado-Gálvez et al., 2018; Aguirre et al., 2018). Scenario-based analysis enables the modeling and simulation of

the complex processes and interactions during and after earthquake events, with a level of detailing that is not possible in a complete probabilistic hazard and risk analysis. As such, the earthquake scenarios are the starting point for such a more detailed risk assessment and for recommendations for improving risk management (e.g., Chatelain et al., 1995; Feliciano et al., 2023).

Representative scenarios are commonly selected based on expert knowledge (e.g., Aguirre et al., 2018) and past events (e.g., Indirli et al., 2011). Synthetic seismic catalogs have also been used for the selection of representative scenarios (McGuire, 1995; Jayaram and Baker, 2009b; Miller and Baker, 2015). A particular approach for scenario selection is based on hazard disaggregation (Bazzurro and Cornell, 1999), which utilizes the conditional probability of different hazard scenarios given an intensity measure (e.g., peak ground acceleration, PGA) at a specific site of interest either equals or exceeds a threshold (Fox et al., 2016; Fox, 2023). As its name suggests, classic hazard disaggregation does not explicitly consider the losses of the affected engineering systems, which are often a function of the intensity measures at multiple locations and which are subject to uncertainty.

The above concepts were extended to loss disaggregation to find earthquake scenarios in terms of magnitude and hypocentral distance that exceed a loss threshold for building portfolios (Goda and Hong, 2009) or infrastructure (Jayaram and Baker, 2009b). Because the spatially accumulated loss can be defined for any portfolio of buildings and infrastructure, loss disaggregation implicitly considers spatially distributed intensity measures. Rosero-Velásquez and Straub (2022) proposed a definition of a representative hazard scenario associated with a loss of return period $t$, e.g., the 100-year loss, which in general does not correspond to the magnitude or intensity measure of the same return period. It is defined as the most likely scenario that leads to the loss value (i.e., its occurrence) associated with this return period $t$. They also presented a numerical procedure for selecting the representative hazard scenario in a continuous space of source parameters with a surrogate model and active learning, thus considering the uncertainty in the conditional losses given a hazard scenario. In the context of seismic risk analysis, earthquake scenarios that are representative of $t$-year loss can be different for different engineering systems, even if they are located in the same area. The definition of Rosero-Velásquez and Straub (2022) differs from the loss disaggregation presented by Goda and Hong (2009) and Jayaram and Baker (2009b), because the latter defines the representative scenario as the most likely one to exceed the $t$-year loss. In this contribution, we compare the two definitions and argue that a definition in terms of the occurrence of the $t$-year loss is more appropriate for most applications, in line with the findings of Fox et al. (2016) for hazard disaggregation.

Considerable work has been devoted to the study of the seismic hazard, vulnerability, and risk in the Valparaíso coastal area of Chile due to its high population density and economic importance in combination with strong seismic activity. Recent earthquakes that led to significant damages occurred in 1971 with $M_w = 7.8$, in 1985 with $M_w = 8.0$ (Indirli et al., 2011), and in 2010 with $M_w = 8.8$ (de la Llera et al., 2017). Recent studies on the Valparaíso area deal with seismic characterization (e.g., Carvajal et al., 2017; Candia et al., 2020), source models (e.g., Poulos et al., 2019; Pagani et al., 2021), ground motion models (Montalva et al., 2017), building exposure models (e.g., Yepes-Estrada et al., 2017; Jiménez et al., 2018; Gómez-Zapata et al., 2022b, b), damage analysis on individual buildings (e.g., Indirli et al., 2011; Jünemann et al., 2015), socio-economic impact (Jiménez Martínez et al., 2020), and seismic risk analysis of the electric power network (Ferrario et al., 2022) and road network (Allen et al., 2022). Additionally, Indirli et al. (2011) identified representative earthquake scenarios using historical

events and expert knowledge, for generating representative ground motion time series, but solely from the hazard point of view and disregarding the risk component.

This paper determines representative earthquake scenarios for different return periods for the residential building stock and the power supply network in Valparaíso and Viña del Mar communes. We adapt and extend the methodology described by Rosero-Velásquez and Straub (2022) for identifying scenarios associated with different return periods from a synthetic earthquake catalog. The representative scenario is found directly by solving a stochastic optimization problem; namely the identification of the mode of the conditional distribution of the source parameters given the occurrence (or exceedance) of the $t$-year loss among the scenarios in the catalog. The stochastic optimization problem is solved with an active learning strategy, whereby the uncertainty in the objective function is estimated by bootstrapping.

We introduce the definition of representative earthquake scenario more formally in Section 2. Then we present the methodology for computing the scenarios on a seismic catalog in Section 3, and illustrate it with idealized examples in Section 4. The description of the study area, the utilized hazard and system models, are presented in Section 5. The results are given in Section 6 and discussed in Section 7. In the paper, we employ the notation and acronyms summarized in Appendix A.

## 2    Definition of representative earthquake scenario

An earthquake scenario can be described by a vector $\theta$ of source parameters, including the magnitude, hypocentral distance, source longitude, latitude and depth. In a stochastic model, the scenario is a single realization of a random vector $\Theta$, with joint probability density function (PDF) $f_\Theta(\theta)$. The PDF of $\Theta$ is obtained from one or more seismic source models (e.g., Poulos et al., 2019) and is conditioned on the occurrence of a seismic event, whose frequency (occurrence rate) is $\lambda_H$.

An earthquake catalog is a set of $n$ earthquake scenarios $\theta^{(1)}, \cdots, \theta^{(n)}$, which are realizations of $\Theta$. The catalog can be a set of synthetic earthquake scenarios, obtained by random sampling from $f_\Theta(\theta)$. Alternatively, the catalog can be obtained from past events (e.g., Poulos et al., 2019).

Synthetic earthquake catalogs have been used in event-based probabilistic seismic hazard analysis (PSHA) and earthquake risk assessment (e.g., Salgado-Gálvez et al., 2018; Ferrario et al., 2022; Allen et al., 2022). The aim of PSHA is to obtain the occurrence rate and distribution of ground motions, taking into account all possible earthquake scenarios (Cornell, 1968; Esteva, 1970); and event-based PSHA utilizes Monte Carlo simulation for sampling earthquake scenarios. Similarly, event-based earthquake risk assessment on spatially distributed systems utilizes synthetic earthquake scenarios for computing the losses, considering the spatial correlation in the ground motion and the vulnerability of the exposed assets (Baker et al., 2021).

For a given earthquake scenario, ground motion models (GMMs) result in spatially distributed intensity measures, e.g., PGA and spectral accelerations, which are input to assess the losses associated with the exposed systems. In the general case, these predictions are stochastic. Thereafter, the model of the engineering system considers the physical and functional vulnerability and results in a loss value $L$.

Because of the randomness and uncertainty in the earthquake scenario, GMM, vulnerabilities and exposure, $L$ is a random variable whose cumulative distribution function (CDF) $F_L(l)$ can be obtained by performing an event-based earthquake risk

assessment for spatially distributed systems with the synthetic earthquake catalog. By combining this CDF with the earthquake occurrence rate $\lambda_H$, one obtains the loss exceedance function $\lambda_L(l)$:

$$\lambda_L(l) = (1 - F_L(l))\,\lambda_H. \tag{1}$$

Based on the loss exceedance function, the losses $l_t$ with a specific return period $t$ can be found as

$$l_t = \lambda_L^{-1}\left(\frac{1}{t}\right) = F_L^{-1}\left(1 - \frac{1}{\lambda_H t}\right), \tag{2}$$

which is defined only for $t \geq \frac{1}{\lambda_H}$. The loss $l_t$ is also called the $t-$year loss.

Following Rosero-Velásquez and Straub (2022), the representative earthquake scenario $\theta_t$, associated with a return period $t$, is defined as the most likely scenario among those causing the $t$-year loss $l_t$. In other words, $\theta_t$ is the mode of the conditional PDF of $\Theta$ given the loss $L = l_t$, also called the loss disaggregation of $\Theta$ given $L = l_t$:

$$\theta_t = \arg\max_\theta f_{\Theta|L}(\theta|l_t). \tag{3}$$

Eq. (3) defines the representative earthquake scenario by conditioning on the occurrence of the loss $l_t$, whereby $l_t$ is defined in terms of exceedance rate. The equation describes the scenario that is most likely to lead to the $t$-year loss $l_t$.

An alternative definition can be formulated in terms of loss exceedance instead of loss occurrence:

$$\theta_t^{exc} = \arg\max_\theta f_{\Theta|L}(\theta|L \geq l_t). \tag{4}$$

Eq. (4) defines the scenario that is most likely to exceed $l_t$. This is the definition corresponding to the classical loss disaggregation (Goda and Hong, 2009; Jayaram and Baker, 2009b). We note that with this definition, in general, the scenario representative of a $t$-year loss will have a return period higher than $t$. Hence, we find its interpretation more difficult, and prefer the definition in Eq. (3). Nevertheless, we propose algorithms to evaluate the representative scenarios according to the two definitions and compare the resulting scenarios in an illustrative example.

Figure 1 illustrates the conditional distributions $f_{\Theta|L}(\theta|l_t)$ and $f_{\Theta|L}(\theta|L \geq l_t)$ for two source parameters $\Theta = (\Theta_1, \Theta_2)$, e.g., representing the magnitude $M_w$ and the hypocentral distance $R$ with respect to a location of interest.

By Bayes' rule, Eq. (3) can be expressed in terms of $f_\Theta(\theta)$,

$$\theta_t = \arg\max_\theta f_{L|\Theta}(l_t|\theta) f_\Theta(\theta), \tag{5}$$

and similarly for the loss exceedance approach:

$$\theta_t^{exc} = \arg\max_\theta \left(1 - F_{L|\Theta}(l_t|\theta)\right) f_\Theta(\theta) \tag{6}$$

wherein $f_{L|\Theta}(l|\theta)$ and $F_{L|\Theta}(l_t|\theta)$ are respectively the conditional PDF and CDF illustrate that the scenario selection criterion balances the probability of the earthquake scenario (quantified by $f_\Theta(\theta)$) and the probability of the $t$-year losses to occur (or being exceeded) at that scenario.

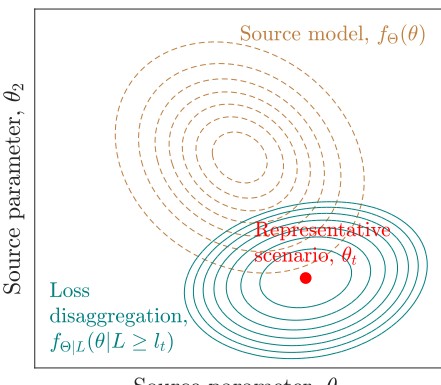

|  (a) Loss occurrence approach | (b) Loss exceedance approach |

**Figure 1.** Illustration of representative scenario in the source parameter space, modified after Rosero-Velásquez and Straub (2022), in terms of loss occurrence (a), and in terms of loss exceedance (b).

To ease the notation in the following section, we let $z_t(\theta)$ denote the objective function of Eq. (5):

$$z_t(\theta) = f_{L|\Theta}(l_t|\theta) f_\Theta(\theta), \tag{7}$$

and $z_t^{exc}(\theta)$ the objective function of Eq. (6):

$$z_t^{exc}(\theta) = \left(1 - F_{L|\Theta}(l_t|\theta)\right) f_\Theta(\theta). \tag{8}$$

## 3  Method for using scenario selection based on a synthetic earthquake catalog

We consider the case where the randomness of earthquake events is represented through a synthetic earthquake catalog. Specifically, we aim at identifying the earthquake scenarios in the catalog that maximize the objective function in Eq. (5) and (6) for different return periods $t$.

The objective functions of Eq. (5) and (6) consist of the PDF $f_\Theta(\theta)$, which is known from the earthquake source model, and the conditional PDF or CDF of $L$ given $\Theta$ evaluated at $l_t$, which can be approximated with conditional samples of losses. To account for the aleatory uncertainty in the modeled ground motions one can draw Monte Carlo samples from the catalog (Silva, 2016) and propagate them to the loss metrics. However, performing this amount of loss evaluations for an entire seismic catalog (normally containing dozens of thousands of events) is computationally (too) expensive. As an alternative, one can use Gaussian process models in combination with active learning to handle aleatory uncertainty more efficiently (Tomar and Burton, 2021; Rosero-Velásquez and Straub, 2022). Furthermore, it has been proposed to pre-select scenarios by the use of extreme value theory and the generalized Pareto distribution (Borzoo et al., 2021).

We propose to first perform only one loss evaluation for each scenario in the catalog and use these to approximate the loss-exceedance function and $l_t$. The same samples are used for an initial approximation of $f_{L|\Theta}(l_t|\theta)$, the second part of the

objective function. This approximation is improved by the use of active learning (AL) to identify earthquake scenarios in the

catalog for which additional loss evaluations are to be performed.

This methodology is an adaptation of the one proposed in Rosero-Velásquez and Straub (2022).

Figure 2 illustrates the main steps of the methodology for selecting representative earthquake scenarios for $n_t$ return periods $t_1 > t_2 ... > t_{n_t}$. The earthquake model in this example is a single seismic source within a bounding volume, and the system is a single building.

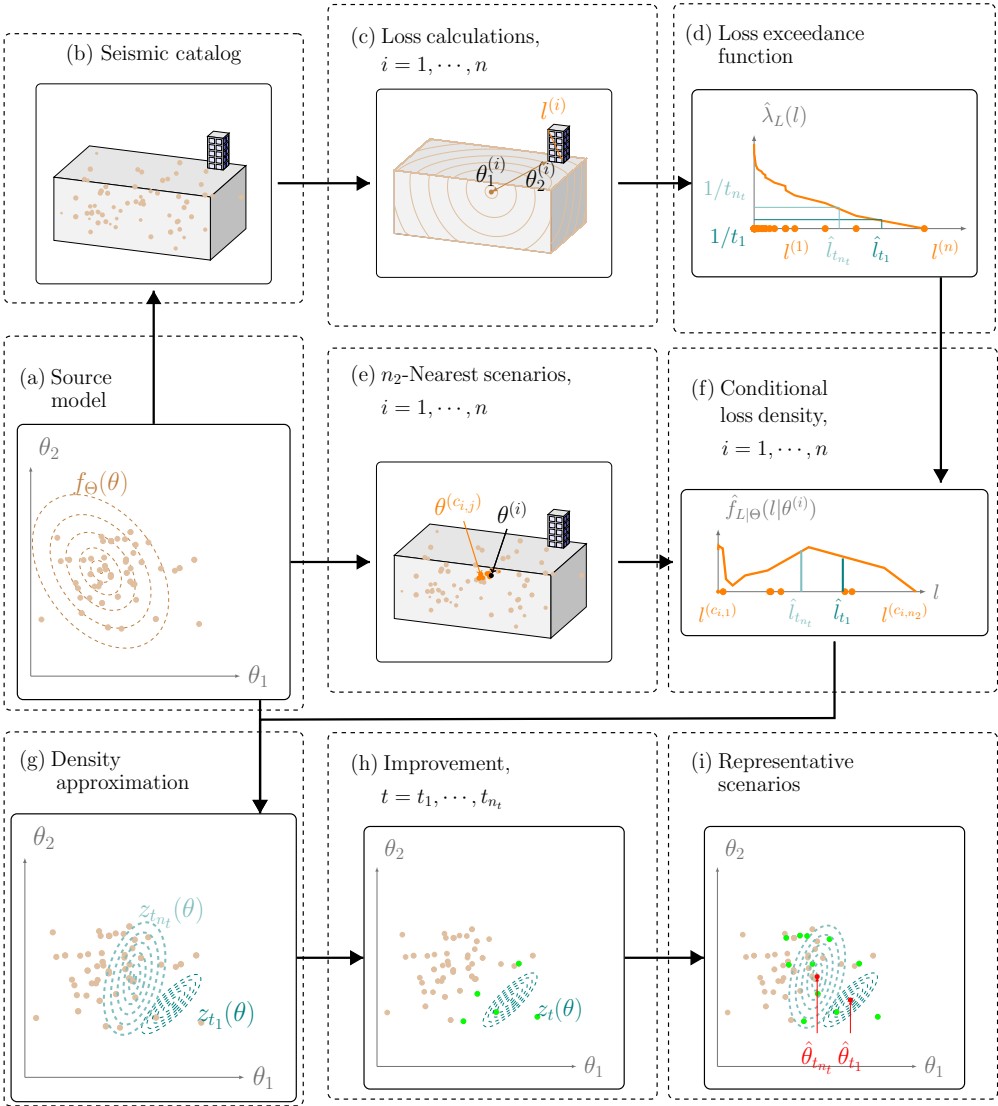

**Figure 2.** General procedure for selecting representative earthquake scenarios with a synthetic earthquake catalog in terms of loss occurrence. Section 3.1 explains with more detail panels (e) and (f), and Sections 3.3 and 3.4 explain panel (h). The remaining panels are referred in Section 3. The procedure in terms of loss exceedance only differs in panel (f), and it is explained in Section 3.2.

The starting point is a seismic source model (panel (a) in Fig. 2), which consists of the occurrence rate $\lambda_H$ and the PDF of the source parameters, $f_\Theta(\theta)$, together with an associated stochastic seismic catalog (panel (b) in Fig. 2). The catalog consists of a set of $n$ random and independent earthquake scenarios $\theta^{(1)}, \cdots, \theta^{(n)}$ generated from $f_\Theta(\theta)$, possibly associated with weights $\omega_1, \cdots, \omega_n$ with $\sum_{i=1}^{n} \omega_i = 1$. The generation of such catalogs for the study site is described in Section 5.2.

For each scenario, one simulates the ground motion fields in terms of the intensity measure (e.g., the peak ground acceleration PGA) through the GMM. These intensity measures are the input to assess the performance of the system components, by combining them with vulnerability functions. Based on the component performances, the total losses in the system, $l$, are evaluated (panel (c) in Fig. 2). Details on the simulation of the ground motion, the system response and loss calculation for the study site are given in Section 5.

From these samples of the system losses, one obtains an estimate of the loss exceedance curve $\hat{\lambda}_L(l)$:

$$\hat{\lambda}_L(l) = \lambda_H \sum_{i=1}^{n} \omega_i \mathbf{1}(l^{(i)} > l), \tag{9}$$

where $\mathbf{1}(\cdot)$ is the indicator function. In addition, one obtains estimates of the $t$-year losses $\hat{l}_t$ for all return periods of interest $t_1, \cdots, t_{n_t}$ following Eq. (2) (panel (d) in Fig. 2).

Since the conditional density of the losses, $f_{L|\Theta}(l|\theta)$, is not available in analytical form, we propose to approximate it with $\hat{f}_{L|\Theta}(l|\theta)$ (panels (e) and (f) in Fig. 2), as detailed in Section 3.1. We utilize this approximation in the objective function of Eq. (5) (panel (g) in Fig. 2) to obtain initial estimates of the objective function $z_t(\theta)$ at each scenario of the catalog, which we denote as $z_t^{(1)}, \cdots, z_t^{(n)}$. To reduce the scatter in the estimates of $z_t$, we add a smoothing step (panel (h) in Fig. 2), which is described in Section 3.3.

At this, and any later stage of the algorithm, we approximate the solution of Eq. (5) by (panel (i) in Fig. 2):

$$i_t = \underset{i=1,\cdots,n}{\arg\max}\, z_t^{(i)} \tag{10}$$

$$\theta_t \approx \theta^{(i_t)} \tag{11}$$

The initial approximation based on a single loss evaluation per scenario in the catalog is typically poor. To enhance the accuracy, we use an active learning strategy (panel (h) in Fig. 2). It intelligently selects earthquake scenarios from the catalog, for which additional loss evaluations are performed. This is presented in Section 3.4.

For the representative earthquake scenario defined by the loss exceedance approach, we approximate the conditional CDF of the losses with an empirical CDF $\hat{F}_{L|\Theta}(l|\theta)$ (analogous to panel (f) in Fig. 2), and utilize this approximation in the objective function Eq. (6) (analogous to panel (g) in Fig. 2) to obtain initial estimates of $z_t^{exc}(\theta)$, denoted by $z_t^{exc(1)}, \cdots, z_t^{exc(n)}$. We then approximate the solution of Eq. (6) by

$$i_t^{exc} = \underset{i=1,\cdots,n}{\arg\max}\, z_t^{exc(i)} \tag{12}$$

$$\theta_t^{exc} \approx \theta^{(i_t^{exc})} \tag{13}$$

## 3.1 Approximation of the objective function $z_t(\theta)$ with kernel density estimation

We approximate the conditional density $f_{L|\Theta}(l|\theta)$ using weighted kernel density estimation (KDE) (Gisbert, 2003). The KDEs at each scenario $\theta^{(i)}$ are evaluated with $n_2$ loss evaluations, which come from the closest scenarios $\theta^{(c_{i,1})}, \cdots, \theta^{(c_{i,n_2})}$ and have associated weights $w_{i,1}, \cdots, w_{i,n_2}$ which sum up to 1, i.e., $\sum_{j=1}^{n_2} w_{i,j} = 1$:

$$\hat{f}_{L|\Theta}(l|\theta^{(i)}) = \sum_{j=1}^{n_2} w_{i,j} \kappa\left(l, l^{(c_{i,j})}, \gamma\right) \tag{14}$$

where $\kappa$ is a kernel function and $\gamma$ is the bandwidth. We define the weights as $w_{i,j} = \exp\left(-d_{i,j}\right) / \sum_{k=1}^{n_2} \exp\left(-d_{i,k}\right)$, where $d_{i,j}$ is the Mahalanobis distance between $\theta^{(i)}$ and $\theta^{(c_{i,j})}$. This ensures that the loss values from scenarios similar to $\theta^{(i)}$ are given more weight in the KDE.

A common choice for $\kappa$ is the Gaussian kernel function , which employs the standard Gaussian PDF $\phi(\cdot)$:

$$\kappa\left(l, l^{(c_{i,j})}, \gamma\right) = \frac{1}{\gamma}\phi\left(\frac{l - l^{(c_{i,j})}}{\gamma}\right) \tag{15}$$

and $\gamma$ computed as suggested in (Silverman, 1986). Alternatively, one can employ a lognormal kernel, excluding the zero loss values (if any), whose probability $p_0^{(i)}$ is estimated from the conditional loss samples $l^{(c_{i,1})}, \cdots, l^{(c_{i,n_2})}$. That is, for $l > 0$ and $l^{(c_{i,j})} > 0$,

$$\kappa\left(l, l^{(c_{i,j})}, \gamma\right) = \left(1 - p_0^{(i)}\right)\frac{1}{l\gamma}\phi\left(\frac{\ln l - \ln l^{(c_{i,j})}}{\gamma}\right) \tag{16}$$

wherein the bandwidth $\gamma$ is computed as suggested in (Silverman, 1986) but only using the logarithm of the nonzero loss samples. In consequence, the weights $w_{i,j}$ have to be adjusted excluding the zero loss samples, i.e.,

$$w_{i,j} = \mathbf{1}\left(l^{(c_{i,j})} > 0\right)\frac{\exp\left(-d_{i,j}\right)}{\sum_{k=1}^{n_2} \mathbf{1}\left(l^{(c_{i,k})} > 0\right)\exp\left(-d_{i,k}\right)} \tag{17}$$

In this case, for scenarios where all the $n_2$ conditional loss samples are zero, the density at $l_t$ equals zero.

The choice of $n_2$ for the KDEs is associated with a trade-off: On the one hand, a small $n_2$ leads to a poor density estimation, but one that is based on loss samples coming from similar scenarios. On the other hand, a large $n_2$ produces a biased KDE from the true conditional density, since it incorporates loss samples of more dissimilar scenarios. However, the bias can be reduced by additional model evaluations. In fact, $n_2$ or more model evaluations at a scenario $\theta$ provide a more accurate KDE than a KDE based on model evaluations coming from the $n_2$ closest scenarios to $\theta$.

For each return period, we obtain an estimate of $f_{L|\Theta}(l_t|\theta^{(i)})$ by evaluating Eq. (14) with argument $\hat{l}_t$. This process is illustrated in panels $e$ and $f$ of Fig. 2. By multiplication with the prior, an estimate of the objective function at all scenarios in the catalog is obtained:

$$z_t^{(i)} = \hat{f}_{L|\Theta}(\hat{l}_t|\theta^{(i)})f_\Theta(\theta^{(i)}) \tag{18}$$

To reduce the significant noise associated with these estimates, we update them with an additional smoothing step described in Sec. 3.3.

Even after the additional smoothing step, the estimates $z_t^{(i)}$ remain subject to uncertainty, due to the limited number of noisy loss evaluations and the need to pool the evaluations from multiple scenarios. For the purpose of the active learning procedure presented in Sec. 3.4, we approximate the uncertainty associated with the objective function values by modeling the estimates $z_t^{(i)}$ as Gaussian random variables $Z_t^{(i)}$. We denote their mean values as $\mu_{Z_t}^{(i)}$ and their standard deviations as $\sigma_{Z_t}^{(i)}$. The mean values are set to $z_t^{(i)}$. We estimate the standard deviations $\sigma_{Z_t}^{(i)}$ for $i = 1, \cdots, n$ via bootstrapping (Efron and Tibshirani, 1993) on their $n_2$ nearest neighbors $n_b$ times, wherein conditional losses are resampled according to the weights $w_{i,j}$.

## 3.2 Approximation of the objective function $z_t^{exc}(\theta)$ with weighted empirical CDF

Eq. (6) contains the conditional CDF of the losses given a scenario. It can be approximated at $\theta^{(i)}$ with a weighted empirical CDF based on the $n_2$ loss evaluations coming from the closest scenarios and their associated weights:

$$\hat{F}_{L|\Theta}(l|\theta^{(i)}) = \sum_{j=1}^{n_2} w_{i,j} \mathbf{1}(l^{(c_{i,j})} < l) \tag{19}$$

The objective function evaluated at scenario $i$ is then estimated as follows:

$$z_t^{exc(i)} = \left(1 - \hat{F}_{L|\Theta}(\hat{l}_t|\theta^{(i)})\right) f_\Theta(\theta^{(i)}) \tag{20}$$

We apply to these estimates the same uncertainty treatment described in Section 3.1 for the KDEs. Thus, we model the estimates as Gaussian random variables $Z_t^{exc(i)}$ with mean $\mu_{Z_t^{exc}}^{(i)} = z_t^{exc(i)}$ and standard deviation $\sigma_{Z_t^{exc}}^{(i)}$ estimated via bootstrapping.

## 3.3 Smoothed estimation of the objective function with Gaussian process regression

To reduce the noise in the estimates of the objective function, we perform an additional smoothing step via Gaussian process regression (GPR) (Rasmussen and Williams, 2005). For each return period $t$ and in each step of the active learning algorithm described in Section 3.4, we perform a separate GPR.

A drawback of GPR is that the computational cost escalates with the size of the training set $n_{train}$. Fitting and estimating the objective function using standard GPR is an $\mathcal{O}(n^3)$ task (Rasmussen and Williams, 2005). Therefore, we perform GPR smoothing only for estimates $\theta^{(i)}$ near the current solution of Eq. (12), and the GPR hyperparameters are learned only once in the first step. Specifically, we identify the $n_{train} = 1500$ nearest scenarios using the Mahalanobis distance, train the GPR and replace the estimate of $z_t^{(i)}$ (resp. $z_t^{exc(i)}$) only for the training set. The other estimates of $z_t^{(i)}$ (resp. $z_t^{exc(i)}$) are left unaltered.

## 3.4 Active learning

An accurate estimation of the objective function is only important near the solution. We exploit this by employing an active learning (AL) strategy to identify scenarios for which further model evaluations are performed.

AL selects scenarios to evaluate through the acquisition function. Here we use the augmented expected improvement (AEI) as an acquisition function (Huang et al., 2006). It approximates for each scenario the expected value of the improvement of the objective function over the current maximum.

We modify the AEI of Huang et al. (2006) with a correction factor, which assesses the quality of the KDE at each scenario. The resulting AEI at scenario $\theta^{(i)}$ is

$$AEI(\theta^{(i)}) = \frac{1}{c_{neigh}^{(i)}} \mathbb{E}\left[\max\left(Z_t^{(i)} - z_t^*, 0\right)\right] \tag{21}$$

wherein $z_t^*$ is the estimate of the objective function at the current best solution $\theta^*$, which is defined as (Huang et al., 2006):

$$\theta^* = \underset{i=1,\cdots,n}{\operatorname{argmax}}\left(z_t^{(i)} + \sigma_{Z_t}^{(i)}\right) \tag{22}$$

The factor $c_{neigh}^{(i)}$ considers the KDE estimation quality at $\theta^{(i)}$. We define it as follows:

$$c_{neigh}^{(i)} = \max\left(\sum_{j=1}^{n_2} \exp\left(-d_{i,j}\right), n^{(i)}\right) \tag{23}$$

wherein $n^{(i)}$ is the sample size of conditional loss values simulated at $\theta^{(i)}$.

The expected value in Eq. (21) is computed in terms of the standard normal PDF $\phi(\cdot)$ and CDF $\Phi(\cdot)$ (Huang et al., 2006):

$$\mathbb{E}\left[\max\left(Z_t^{(i)} - z_t^*, 0\right)\right] = \left(\mu_{Z_t}^{(i)} - z_t^*\right)\Phi\left(\frac{\mu_{Z_t}^{(i)} - z_t^*}{\sigma_{Z_t}^{(i)}}\right) + \sigma_{Z_t}^{(i)}\phi\left(\frac{\mu_{Z_t}^{(i)} - z_t^*}{\sigma_{Z_t}^{(i)}}\right) \tag{24}$$

For each return period $t$, we perform $n_l$ loss evaluations at the $n_s$ scenarios with the largest $AEI$. Taking into account the $n_s \times n_l \times n_t$ new model evaluations, we update the KDEs, the density observations $z_t^{(i)}$ and the bootstrap standard deviations $\sigma_{Z_t}^{(i)}$. At scenarios where more than $n_2$ loss evaluations have been computed, we deviate from Eq. (14) and evaluate the KDE with all these evaluations (instead of only $n_2$ evaluations).

The AL steps are repeated until convergence is achieved or the maximum number of AL iterations $n_3$ is exceeded. Convergence is achieved when the $AEI$ of all scenarios is below a threshold $\epsilon$ for at least $n_d$ consecutive AL iterations, which prevents premature stopping. A suggested value for $n_d$ is $d+1$, wherein $d$ is the dimensionality of the source parameter random vector $\Theta$ (Huang et al., 2006). We also choose $n_3 = 1000$ for encouraging the AL procedure to stop by convergence. The threshold $\epsilon$ is chosen as (Huang et al., 2006):

$$\epsilon = r \times \left(\max_{i=1,\cdots,n} z_t^{0,(i)} - \min_{i=1,\cdots,n} z_t^{0,(i)}\right) \tag{25}$$

$z_t^{0,(i)}$ is the initial evaluation of the objective function, which is computed before the AL.

An analogous derivation of the $AEI$ is obtained for the case of the objective function in terms of loss exceedance, i.e. $z_t^{exc(i)}$.

## 4 Illustrative examples

In this section, we present two simple examples to illustrate the methodology. The first one is a one-dimensional example, where we focus on the performance of AL and the approximations of the objective function obtained with the noisy KDE estimations and GPR. The second one is a two-dimensional example, where we show the variability of the scenario selection and the solutions computed with the loss ocurrence and exceedance approaches. In both examples, the exact solution is known.

## 4.1 Building portfolio subjected to a single seismic source with variable magnitude

Figure 3 illustrates the performance of the acquisition function on a one-dimensional example. In the example, the only source parameter is the magnitude $M_w$, which is beta distributed with shape parameters $\alpha = 1$, $\beta = 3$, and scaled between 0 and 10. The conditional distribution of the logarithm of the losses associated with the building portfolio given $M_w = m_w$ is a normal random variable with mean $\mu_{\ln L}(m_w) = -\frac{1}{2}\sin(\frac{5}{2}(m_w - 5)) + \exp(\frac{m_w - 5}{7}) + 7$ and standard deviation $\sigma_{\ln L} = 0.7$. We set $l_t = 10$ for this example.

We generate a synthetic earthquake catalog of size $n = 100$. The KDEs are computed with Gaussian kernel and $n_2 = 70$ loss samples, or more if the scenario has more than $n_2$ conditional loss evaluations, and the bootstrap standard deviation is computed based on $n_b = 100$ samples. We perform the GPR on the whole catalog and learn the hyperparameters at every AL step, since the catalog size in this example is not restrictive. For the AL stage, we select $n_s = 5$ scenarios per AL iteration for computing $n_l = 10$ loss evaluations at each scenario (i.e., $n_s \times n_l = 50$ damage evaluations per AL iteration). The acquisition function is the $AEI$, as introduced in Eq. (21). We also let the algorithm to achieve convergence with a maximum of $n_3 = 1000$ AL iterations, with the convergence criterion in Eq. (25) and $r = 0.001$.

Figure 3 compares intermediate and final results for this example to the true results. After the initial loss evaluations at the 100 scenarios, the estimate of the objective function is poor. However, the acquisition function is able to select scenarios near the true solution. In the final step, one can observe that estimates of the objective function values have high noise, but the GPR is effective in reducing this noise. The resulting estimate of the objective function is close to the true value around the optimum.

## 4.2 Building portfolio subjected to an earthquake with unknown magnitude and location

This simple example is adapted from Rosero-Velásquez and Straub (2022). It considers a hypothetical fault, where strong earthquakes occur with a rate of $\lambda_H = 0.3 \, \text{yr}^{-1}$. We consider the damages that earthquakes cause to a building portfolio in a small town. The source parameters $\Theta = [M_w, \ln R]^\mathsf{T}$ are the magnitude $M_w$ and the average hypocentral distance $R$ from the earthquake source to the buildings. The source model for $\Theta$, $f_\Theta(\theta)$, is a normal distribution with mean vector $\mu_\Theta$ and covariance matrix $\Sigma_\Theta$ given as follows:

$$\mu_\Theta = \begin{bmatrix} 7.00 \\ 4.38 \end{bmatrix}, \quad \Sigma_\Theta = \begin{bmatrix} 0.36 & -0.08 \\ -0.08 & 0.49 \end{bmatrix}$$

A standard deviation $\sigma$ represents the uncertainty in the ground motion, damage measure, and losses. The losses $L$ are a log-normal random variable with parameters $\mu_{\ln L} = -3.16$, $\sigma_{\ln L} = \sqrt{2.46 + \sigma^2}$. With these choices, the conditional density $f_{\Theta|L}(\theta|l_t)$ can be evaluated analytically. It is a normal distribution, whose mean vector is the representative earthquake scenario for a return period $t$.

We set $\sigma = 0.5$, and use return periods of $50 \, \text{yr}$, $100 \, \text{yr}$, $500 \, \text{yr}$ and $1000 \, \text{yr}$. The resulting exact representative earthquake scenarios with the loss occurence approach are: $\theta_{50} = [7.42, 3.42]^\mathsf{T}$, $\theta_{100} = [7.51, 3.21]^\mathsf{T}$, $\theta_{500} = [7.69, 2.81]^\mathsf{T}$, and $\theta_{1000} = [7.75, 2.65]^\mathsf{T}$. We use them to verify the proposed sampling-based algorithm.

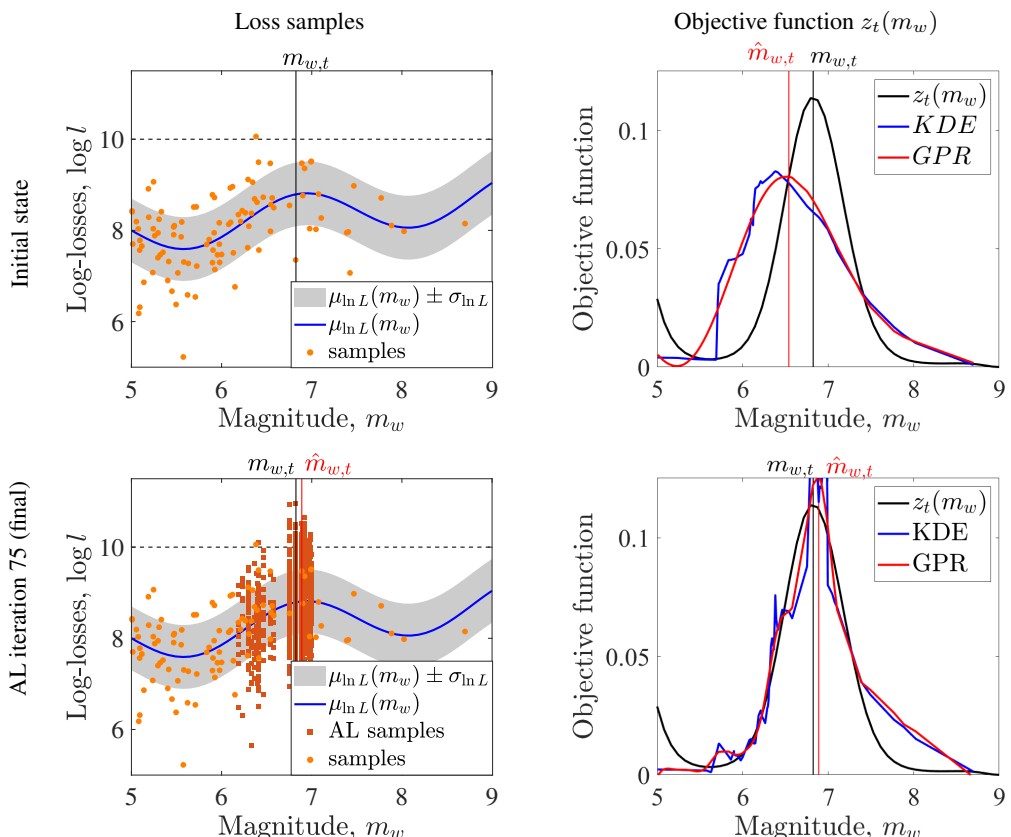

**Figure 3.** Illustration of active learning (AL) for maximizing the objective function $z_t(m_w)$ with fixed value $l_t$, marked with dashed line in the plots in the left panel. The solution $m_{w,t}$ is approximated with the sample point $\hat{m}_{w,t}$. The loss samples before and during the AL steps are shown on the left panel. The approximations of the objective function, either with kernel density estimation (KDE) or Gaussian process regression (GPR), are shown in the right panel.

To estimate $\theta_t$ with the proposed methodology, a synthetic earthquake catalog with $n = 2 \times 10^4$ random scenarios is employed. We simulate the losses once at each scenario, approximate $l_t$, and compute the KDEs at each scenario. The KDEs are based on $n_2 = 200$ loss values, computed with Gaussian kernel, and the bootstrap variance with $n_b = 100$ repetitions. We perform GPR with a training set of size $n_{train} = 1500$, which is constructed as described in Section 3.3.

The maximum number of AL iterations is $n_3 = 1000$, where at every step the losses are evaluated at $n_s = 2$ scenarios $n_l = 10$ times, for each return period. The procedure stops after the maximum $AEI$ is below $\epsilon$, with $r = 0.001$, for at least $n_d = 5$ consecutive AL iterations. For analyzing the uncertainty in the estimation of $\theta_t$, we repeat the experiment 20 times. For the $50, 100$ and $500$-year representative scenarios, all experiments converged in less than 10 AL iterations, whereas for the 1000-year return period at most 30 AL iterations were required. For the loss exceedance approach, fewer iterations were required in general.

Figure 4 shows the resulting representative hazard scenarios for each return period and their spread, which is mainly caused by the numerical approximation of the objective function with limited number of samples. As expected, one can observe that the representative scenarios are more extreme when using the loss exceedance approach.

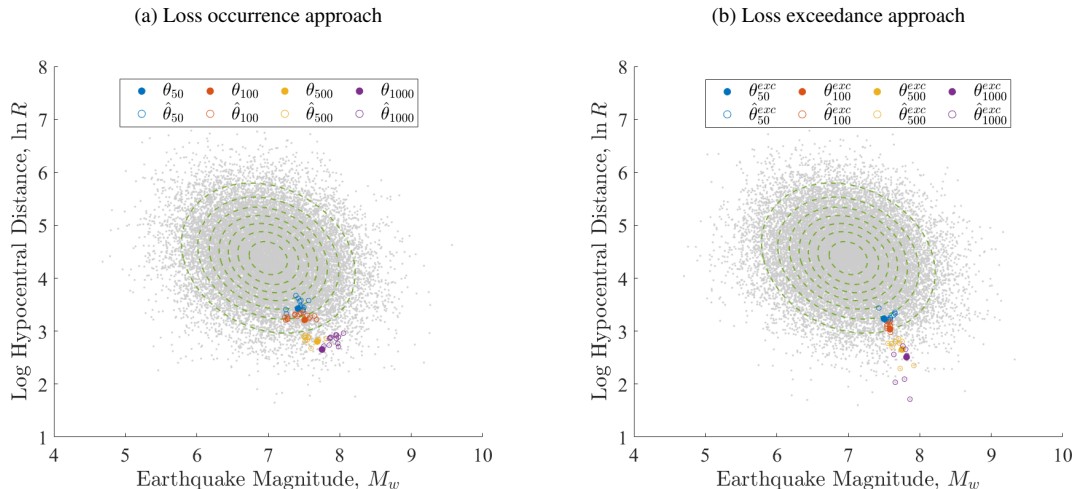

**Figure 4.** Numerical approximation of the representative earthquake scenarios. Panel (a) shows the representative scenarios computed with the loss occurrence approach, $\hat{\theta}_t$, and panel (b) those computed with the loss exceedance approach, $\hat{\theta}_t^{exc}$. The representative earthquake scenarios correspond to four different return periods $t = 50, 100, 500, 1000$ years, based on a Monte Carlo sample of scenarios. Each return period is represented by a different color. For each return period, the 20 approximations $\hat{\theta}_t$ (resp. $\hat{\theta}_t^{exc}$), corresponding to 20 experiments, are the colored empty circles, and the corresponding exact solutions are depicted by filled circles. The grey points are the scenarios of the catalog, and the dashed contours represent the probability density function (PDF) of the source parameters.

## 5 Case study: Valparaíso and Viña del Mar communes

### 5.1 Context of the study area

We apply the proposed methodology to determine representative earthquake scenarios for the communes of Valparaíso and Viña del Mar, which are located in the Valparaíso Region of Chile on the Pacific coast. The study area is the second-largest Chilean urban centre; based on the latest Chilean census (INE, 2017), it is home to 630 903 inhabitants. It hosts the Port of Valparaíso, which is an important container and the main passenger port in Chile. The area shows a heterogeneous building inventory, ranging from apartment buildings to informal settlements, and a historic district declared a World Heritage Site by UNESCO in 2003 (Indirli et al., 2011; Jiménez et al., 2018).

The National Electric System (SEN) provides the area's power supply. The SEN is the largest Chilean transmission grid, and covers most of the national territory. The SEN connects power plants and substations with the consumer areas through

high-voltage lines. The topology of the SEN is characterized as a single-scale network with a fast decaying tail, and most of the load substations are close to a generation unit, with a median distance of 9 km (Ferrario et al., 2022).

Powerful earthquakes have hit the area in the past, such as the 1730 earthquake, with inferred magnitude $M_w$ in the range 9.1—9.3, and the 1906 event, with inferred moment magnitude $M_w$ 8.0—8.2 (Carvajal et al., 2017). More recently, the 1985 $M_w$ 8.0 event affected around 230 000 dwellings, 1 million people, affecting the Regions of Valparaíso, Maule, O'Higgins and Metropolitan Region of Santiago de Chile, and caused losses in residential buildings of about 1.4 billion USD [in 1985 values] (ONEMI, 1985). The most recent $M_w$ 8.8 Maule earthquake (2010) caused severe structural damage in buildings in Viña del Mar, including in buildings retrofitted in 1985 (Jünemann et al., 2015).

A hazard evaluation of the Valparaíso urban area presented by Indirli et al. (2011) selected representative earthquake scenarios based on historical events, considering the seismicity around the study area, to specify an average regional seismic input and to generate synthetic seismograms. The scenarios are summarized in Table 1; their magnitudes range from $M_w = 5.7$ to $M_w = 8.2$.

**Table 1.** Representative earthquake scenarios for the urban area of Valparaíso selected by Indirli et al. (2011) from historic events. The epicenter location is reported with a map by Indirli et al. (2011), hence their numeric values, as well as moment magnitude and depth, are here reproduced from the earthquake records of the USGS ComCat Catalog (USGS, 2023). The 1985 $M_w$ 8.0 earthquake event (lon = $71°51'$W, lat = $33°14'$S, depth = 33 km) has similar source parameters as the 1906 event, while the 2010 earthquake event occurred after the study of Indirli et al. (2011) was submitted for publication. The event dates are in local time. The locations of the epicenters are shown in the map of Figure 11.

| Source param., $\theta$ | Event date | | | |
|---|---|---|---|---|
| | 16/08/1906 | 28/03/1965 | 06/07/1979 | 16/10/1981 |
| Longitude | $72°24'$W | $71°13'$W | $71°19'$W | $73°4'$W |
| Latitude | $32°24'$S | $32°31'$S | $32°9'$S | $33°8'$S |
| Depth [km] | 35 | 70 | 45 | 33 |
| Magnitude, $M_w$ | 8.2 | 7.4 | 5.7 | 7.2 |

## 5.2 Earthquake model and synthetic earthquake catalog

We employ the earthquake model presented by Poulos et al. (2019) to generate a synthetic catalog of earthquake scenarios. The catalog has $2 \times 10^4$ scenarios with magnitude larger than or equal to $M_w = 5.0$, which is the minimum magnitude defined by Poulos et al. (2019) for performing the declustering on the historical seismic catalogs on which the earthquake model is based on. The catalog covers the whole country of Chile, and consists of scenarios at the subduction interface and subduction intraslab zones. The earthquake model utilizes the slab geometry proposed by Hayes et al. (2012) for the depth contours and trench geometry, and divides the Chilean subduction zone into three subduction interface and four intra-slab zones, whose combined occurrence rate equals $\lambda_H = 43.3 \text{ yr}^{-1}$.

The epicentral locations of the catalog are generated randomly, based on the occurrence rate associated with the seismic zones defined by the occurrence model. The magnitude is sampled with an importance sampling (IS) approach. We employ a uniform distribution with minimum and maximum values defined by the magnitude range of each seismic zone, as IS density. The corresponding IS weights are considered when determining the loss-exceedance function and computed following the original Gutenberg-Richter relationship at each seismic zone (Poulos et al., 2019). The resulting catalog is depicted in Figure 5, in which one can observe that events of different magnitudes have similar spread within the seven seismic zones.

The earthquake model of Poulos et al. (2019) only considers the subduction zone, hence the independent source parameters are the moment magnitude $M_w$, longitude $X$ and latitude $Y$ of the epicenter. Other parameters, such as the depth $H$, strike, dip and rake angles, are determined by the geometry derived by Hayes et al. (2012), depending of the epicenter location. Therefore, the PDF of the source parameters $f_\Theta(\theta)$ is represented, respectively, by the conditional PDF $f_{M_w|X,Y}(m_w|x,y)$, and the location-dependent occurrence rate $\lambda(x,y)$:

$$f_\Theta(\theta) \propto \lambda(x,y) f_{M_w|X,Y}(m_w|x,y) \tag{26}$$

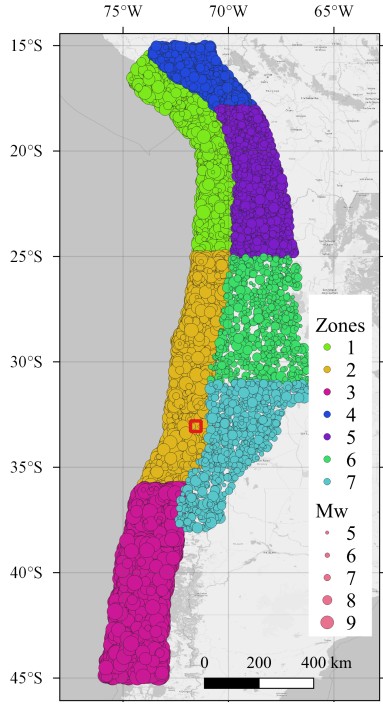

**Figure 5.** Synthetic earthquake catalog with 20 000 scenarios (Poulos et al., 2019). The circle size corresponds to the scenario magnitude. The red square contains the study area. Seismic zones 1 to 3 are of subduction interface type, and zones 4 to 7 are of subduction intra-slab type. Basemap from ©OpenStreetMap contributors 2023. Distributed under the Open Data Commons Open Database License (ODbL) v1.0.

### 5.2.1 Ground motion models

For the residential building stock, we evaluate the PGA and spectral accelerations at $0.3\,\text{s}$ and $1.0\,\text{s}$ with the Ground Motion Model (GMM) presented by Montalva et al. (2017). The uncertainty in the median prediction is modelled with a Gaussian random field, with the spatial correlation model of Jayaram and Baker (2009a). The choice of these models is based on the epistemic uncertainty analysis of different ground motion and correlation models by Gómez-Zapata et al. (2022a), who analyzed the same study area. For the scope of this study, we do not consider cross-correlated ground motion fields.

For the Chilean power network, we employ the GMM of Abrahamson et al. (2016) and the spatial correlation model developed by Goda and Atkinson (2010) for predicting the PGA. This is the same ground motion model as the one utilized by Ferrario et al. (2022).

The functional form of both GMMs is similar, and therefore, their predictions do not differ significantly, as observed in previous studies (e.g., Hussain et al., 2020; Gómez-Zapata et al., 2022a). In particular, Hussain et al. (2020) found negligible differences in direct loss estimates for the residential building stock of Santiago de Chile after using these two GMMs to simulate the associated ground motion from subduction earthquake scenarios.

### 5.3 Model for the building stock in the communes of Valparaíso and Viña del Mar

We employ the Bayesian exposure model of the building stock with the building classes described in (Gómez-Zapata et al., 2022a) and available in (Pittore et al., 2021). The model was constructed by taking the OpenStreetMap footprint of the buildings in the two communes, and assigning to each footprint the most likely building class. The buildings are counted within a regular $500\,\text{m} \times 500\,\text{m}$ resolution grid in the urban areas, as shown in Fig. 6. Detailed building counts for each class are presented in (Gómez-Zapata et al., 2022a)

The model considers 16 building classes (Gómez-Zapata et al., 2022a), which correspond to the ones proposed in the SARA project (Yepes-Estrada et al., 2017), and have an associated replacement cost. Furthermore, each building class has an associated fragility model with five damage states (Villar-Vega et al., 2017). The fragility model for each building associates an intensity measure (spectral acceleration at $0.3\,\text{s}$, $1.0\,\text{s}$, or the PGA) with the probabilities of achieving a damage state. We assume the following relative replacement cost percentages for each damage level: $0\%$ for no damage, $2\%$ for slight damage, $10\%$ for moderate damage, $50\%$ for extensive, and $100\%$ for complete damage.

We utilize the model to evaluate the ground motion and simulate the building damage. Given an intensity level of the ground motion, the damage is simulated randomly at each building with a discrete distribution with probabilities defined by the fragility functions. The losses for each scenario in the catalog are computed as the accumulated reconstruction cost of the damaged residential buildings, based on the simulated damage.

### 5.4 Model for the Chilean National Electric System, SEN

We model the SEN and its components, following Ferrario et al. (2022). The network model consists of 1494 nodes, representing 500 generation units and 994 substations, and the transmission lines connecting them, with a total power generation

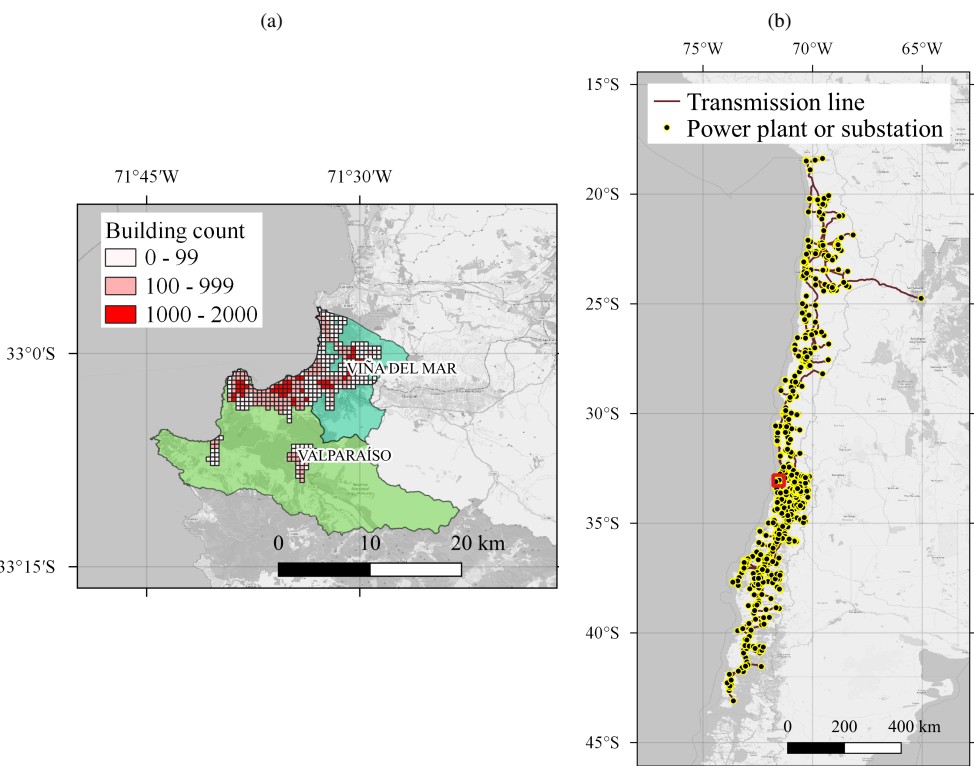

**Figure 6.** (a) Exposure model of the residential building stock. Each rectangular cell shows the total count of residential buildings, indicating the most dense areas. Source: Pittore et al. (2021). (b) Geographic location of the National Electric System (SEN). The network follows the narrow shape of the country, and the communes of Valparaíso and Viña del Mar (inside red square) are in a central location within the network. Source: Coordinador Eléctrico Nacional, version 2019. Basemap from ©OpenStreetMap contributors 2023. Distributed under the Open Data Commons Open Database License (ODbL) v1.0.

capacity of 21.9 GW. The model considers seismic interaction and system performance subjected to component failures. Given a scenario with a ground motion field (in this case, the PGA), each node is randomly associated with a damage state, by means
of the fragility function, and a recovery time associated with their damage state.

The losses associated with the SEN due to an earthquake scenario are quantified in terms of the energy not supplied (ENS). The ENS is evaluated at each substation by solving the power in normal steady state operation through the Direct Current - Optimal Power Flow (DCOPF) model (Wood et al., 2013), and comparing it with the power in a damaged state operation caused by the earthquake scenario. To quantify the loss in the power supply in the communes of Valparaíso and Viña del Mar,
we calculate the total ENS with the sum of the ENS of all substations located in the two communes (14 in total).

DCOPF is typically adopted in practice for transmission networks (Frank and Rebennack, 2016). It optimizes the power generation cost, taking into account the capacity of the power plants and transmission lines connected to the power grid, the

generation cost associated with each power plant, and the demand from the clients. For modeling the system response to an earthquake, the DCOPF considers the reduced capacity of components affected by the earthquake. A detailed description and validation of the network model of the SEN can be found in (Ferrario et al., 2022).

## 6 Evaluation of representative earthquake scenarios for Valparaíso and Viña del Mar communes

### 6.1 Results for the residential building stock

Figure 7 shows the annual exceedance rate of the losses [in 2016 USD], based on the replacement costs estimated by Yepes-Estrada et al. (2017). The USGS ComCat Catalog records that between 1960 and 2020 there were 12 seismic events that produced a macroseismic intensity greater or equal than VI on the Mercalli scale in the two communes (USGS, 2023). This corresponds to an occurrence rate of $0.20\,\mathrm{yr}^{-1}$. It is reasonable to assume that events with macroseismic intensity of VI or higher lead to losses of at least 10 million USD. For comparison, the reparation cost of the residential buildings in the two communes due to the 1985 earthquake event was USD 49.7 million [in 1985 values, which corresponds to 110.9 million USD in 2016] (ODEPLAN, 1985). This data therefore validates the lower end of the loss exceedance rate obtained with the synthetic earthquake catalog.

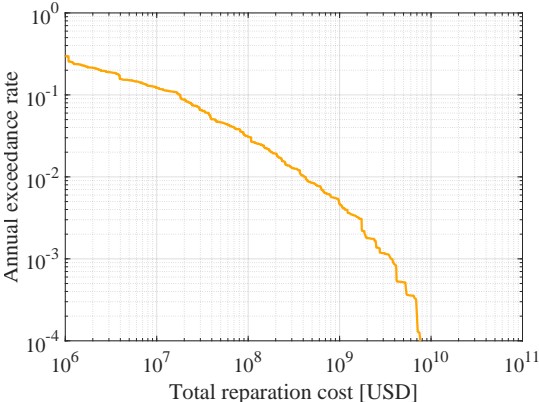

**Figure 7.** Loss-exceedance function of the reparation costs [USD in 2016 values] associated with the residential building stock in Valparaíso y Viña del Mar communes.

As in Section 4.2, we evaluate the representative scenarios in 20 independent runs of the algorithm, to check the robustness of the results. In all evaluations, we found a spread of the identified representative scenarios, similar to that of Figure 4. This spread is larger for higher return periods, but most of the numerical solutions (11 out of 20 for the 1000-year loss return period and at least 16 out of 20 for the other loss return periods) have epicentral location within a radius of $50\,\mathrm{km}$ around the mode, and the coefficient of variation of the magnitude is below $4\%$ for all return periods. In the following, we only present the modes, i.e., the representative scenarios that were identified the most frequently in the 20 repetitions.

Figure 8 shows the representative earthquake scenarios for the analyzed return periods with loss occurrence approach. One can observe that large return periods are associated with scenarios that have larger magnitude. The fact that the magnitude for the 1000-year scenario equals only $M_w = 7.01$ is a consequence of the size of the study area. On the one hand, among the 36 seismic events with $M_w \geq 8.0$ registered along the Chilean coast between 1570 and 2023 (CSN, 2023), only 4 events had an epicenter near the two communes, i.e., within a radius of approximately 70 km. For comparison, the identified 1000-year scenario is at a distance of around 20 km from Valparaíso and Viña del Mar. On the other hand, the spatial correlation of the ground motion within a small study area leads to an increased likelihood of extreme losses in a scenario with lower earthquake magnitude and extreme ground motion residuals. This tendency was also found by Goda and Hong (2009) with the loss exceedance approach.

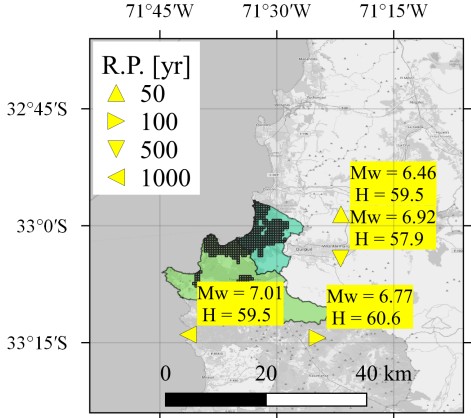

**Figure 8.** Representative earthquake scenarios for the residential building portfolio in Valparaíso and Viña del Mar communes for four different return periods (RP). The hypocentral depth $H$ is displayed in km. Source of the exposure model: Pittore et al. (2021). Basemap from ©OpenStreetMap contributors 2023. Distributed under the Open Data Commons Open Database License (ODbL) v1.0.

## 6.2 Results for the power network

Figure 9 shows the loss exceedance function in terms of the ENS obtained with the synthetic earthquake catalog. The largest sampled ENS value is around $2 \times 10^5$ MWh, which is around 20% of the annual energy demand of the two communes.

The spread in epicentral locations of the representative scenarios obtained with the 20 runs is larger than the one of the residential building stock, but is still small. At least 13 solutions cluster around the sample mode within a radius of 100 km, and the coefficient of variation of the magnitude is below 5% for all return periods.

Figure 10 shows the resulting representative earthquake scenarios for the analyzed return periods. One can observe that the scenarios are close to the two communes but less concentrated than those of the residential building stock, and have a different magnitude range. This reflects the fact that the total ENS, although computed only at the substations located within the two communes, depends on the damage state of the components of the rest of network.

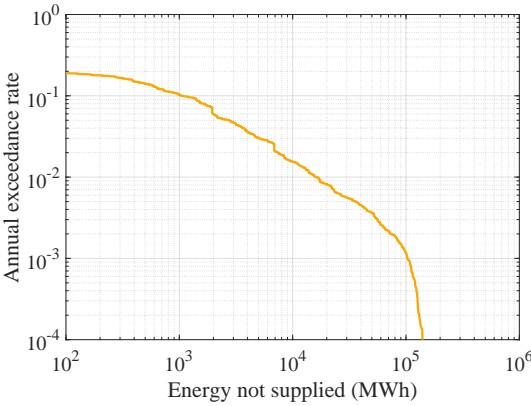

**Figure 9.** Loss-exceedance function of the total energy in megawatt hour (MWh) not supplied in communes of Valparaíso and Viña del Mar.

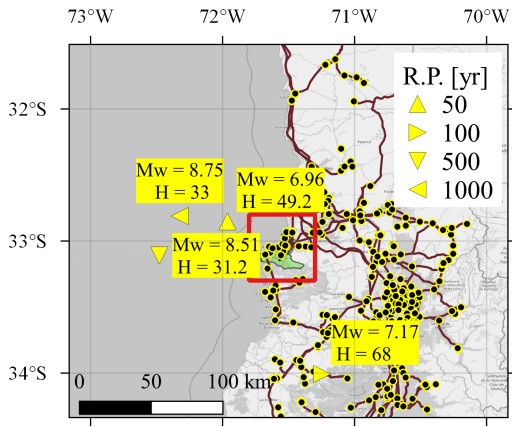

**Figure 10.** Representative earthquake scenarios for the power supply considering the total energy not supplied (ENS) of Valparaíso and Viña del Mar communes for four different return periods (RP). The red square indicates the location of the two communes, and the hypocentral depth $H$ is displayed in km. Source of the power network: Coordinador Eléctrico Nacional. Basemap from ©OpenStreetMap contributors 2023. Distributed under the Open Data Commons Open Database License (ODbL) v1.0.

Even though the power supply network is spread out over a larger area, the representative earthquake scenarios are close to the study area. They reflect that the most important components of the network for the two communes are in their proximity. For example, the source location of the 100-year return period scenario lies near a main connection between the substations in the two communes and the rest of the SEN.

### 6.3 Comparison with past earthquake events

Figure 11 compares the results with the historical events selected in Indirli et al. (2011). Although the representative earthquake scenarios and the selected historical events target the same area of interest, they have different purposes. The historical events

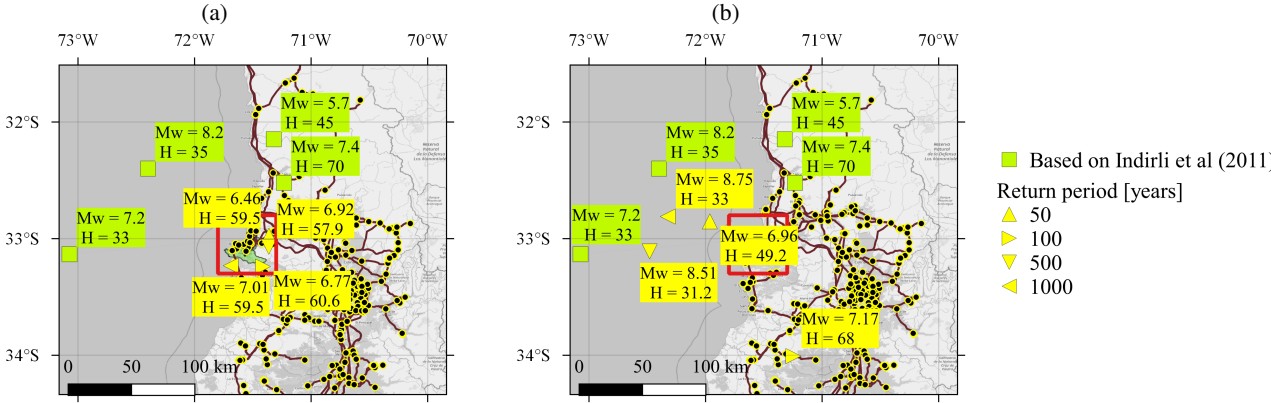

**Figure 11.** Representative earthquake scenarios for the (a) building stock and (b) power supply, compared with past earthquake events selected in (Indirli et al., 2011). The location of the two communes is within the red square, and the National Electric System (SEN) network is also displayed. Source of the exposure model: Pittore et al. (2021). Source of the power network: Coordinador Eléctrico Nacional. Basemap from ©OpenStreetMap contributors 2023. Distributed under the Open Data Commons Open Database License (ODbL) v1.0.

presented by Indirli et al. (2011) aim at representing the seismicity of the most important seismic zones affecting the study area. In contrast, the representative earthquake scenarios, as defined in Rosero-Velásquez and Straub (2022), take into account

the performance and the losses caused by damage and failures in the analyzed engineering system. In addition, the scenarios are selected based on different loss levels, which are attached to return periods.

### 6.4 Computational cost

In terms of loss evaluations, the analysis requires one evaluation per scenario in the catalog for constructing the loss exceedance function with event-based earthquake risk assessment. That corresponds to $2 \times 10^4$ loss evaluations. In addition, during the AL

stage, around 10 iterations were necessary to achieve the convergence criterion of Eq. (25), each of them consisting of 160 new loss evaluations ($n_s = 2$ scenarios evaluated $n_l = 20$ times, for each of the $n_t = 4$ return periods). Therefore, 1600 loss evaluations are needed to find the representative earthquake scenarios for 4 different return periods.

For comparison, Goda and Hong (2009) report that they use a total of $5 \times 10^6$ loss evaluations for the classical loss disaggregation. Furthermore, they only evaluate the scenarios with the loss exceedance approach. Extending the loss disaggregation

approach to loss occurrence will likely require additional evaluations. Additionally, the computation cost of the loss disaggregation approach scales exponentially with the number of parameters describing seismic scenarios. Hence, the classical loss disaggregation approach will not be applicable to problems in which earthquake scenarios are described by more than 3 or 4 parameters. By contrast, we successfully tested the proposed approach for seismic hazard models with 7 parameters in applications not reported in this paper.

## 7 Discussion

### 7.1 On the results for the communes of Valparaíso and Viña del Mar

The representative earthquake scenarios summarized in Fig. 11 can provide important input to risk assessment and risk management activities. The fact that the scenarios identified with the proposed approach differ from the historical events selected in Indirli et al. (2011) should not be surprising, as the latter are in some sense just "random samples" of earthquake events. Nevertheless, the historic events can provide a useful validation of the identified scenarios. In this regard, the scenarios identified as representative of the power supply network appear to be in line with the historic events. The identified 500 and 1000-year scenarios have larger magnitudes than the historical events, which is expected since the historical events come from a (roughly) 100-year period, as shown in Table 1. By contrast, the representative scenarios identified for the building stock have smaller magnitudes than the historical events. However, they occur much closer to the considered building stock. Furthermore, according to the employed model, extreme losses are more likely to occur by a combination of a less strong earthquake with larger-than-average ground motions (i.e., a large value of the inter-event term in the GMM). This effect occurs for the residential building stock due to its spatial concentration and not (or to a much smaller extent) for the power supply network, which is spatially distributed.

The above observations lead to some important conclusions: Firstly, the scenarios, rightly, are different for different assets. Secondly, the scenarios depend on model assumptions beyond the seismic source models. In the application here, the model of the ground motion variability has a distinct effect on the scenarios for the residential building stock. Given that the employed state-of-the-art model might overestimate the event-to-event variability (Bodenmann et al., 2023), the results for the building stock should be utilized carefully. Thirdly, because the earthquake scenario in the building case is representative of certain loss return periods only in combination with high ground motions, it should be investigated if and how the representative scenario should also provide ground motion fields together with the earthquake event. We note that these issues are also present for the classical hazard and loss disaggregation methods. Overall, for practical risk management tasks it is recommended to use the historic events jointly with the identified scenarios, in particular for the residential building stock case.

The dependence of the results on the engineering system and the model assumptions implies that the representative scenarios should be regularly updated, depending on how much the analyzed system changes and the models improve. According to demographic projections of the INE, based on the 2017 Chilean Census, the population in the two communes will increase by around 13% by 2035 with respect to the population in 2017 (INE, 2017). This demographical change will likely cause changes in the residential building stock and the power demand. Depending on how these changes develop, the representative earthquake scenarios may change as well. Furthermore, the presented results do not consider the potential impact caused by further earthquake-triggered hazards, such as tsunamis and landslides. The tsunami impact during offshore earthquake scenarios with large magnitude should be considered in a complete loss estimation, and may affect the scenario selection associated with large return periods. Similarly, the scenario selection may change if one considers the landslides potential in the study area. Finally, cascading effects have not been considered in the power network model. Although the topology of the SEN and the redundancy of generators along the network reduce the probability of large blackouts, hub nodes far from the two

communes may affect them through a sequence of failures triggered by earthquake scenarios impacting them. This may shift the representative scenarios to further locations from the study area.

## 7.2 On the definition of representative scenarios and the methodology for identifying them

We presented the evaluation of representative earthquake scenarios based on the loss occurrence and the loss exceedance approach; the latter coincides with the classical loss disaggregation method (Goda and Hong, 2009; Jayaram and Baker, 2009b). In the illustrative example of Section 4.2, we compare the results of the two approaches. For the case of hazard disaggregation, it has been proposed in the literature that the results of both approaches should be reported (Fox, 2023). However, we decided against reporting the scenarios of the exceedance approach for the Valparaíso and Viña del Mar communes, to avoid confusion. We find the loss occurrence approach to have a more intuitive interpretation. Scenarios identified with this approach correspond to a loss that is the $t$-year loss, which can be reported jointly with the scenarios. They are the most likely scenarios leading to this value (which on average is exceeded once in $t$ years). By contrast, we find it difficult to communicate the meaning of the scenarios with the loss exceedance approach – and we believe it will be mostly misunderstood. Scenarios obtained with the loss occurrence approach can be described as "representative of a loss that is exceeded on average once in $t$ years". For the loss exceedance approach, one would need to describe scenarios as "representative of the losses that would occur when conditioning on a loss at least as large as the one that would be exceeded once in $t$ years", which seems too convoluted to communicate effectively. We also have difficulties to conceive of a risk management activity for which such a definition would be more appropriate. However, we acknowledge that this discussion could benefit from additional comparisons of the two approaches in future studies.

To evaluate the representative scenarios, we adapted the methodology of Rosero-Velásquez and Straub (2022). The methodology leads to lower computational cost in terms of loss evaluations compared to the classical loss disaggregation. By incorporating active learning, the methodology concentrates the conditional loss evaluations around the scenarios that most likely produce the $t$-year loss value $l_t$. This concentration of samples around the solution and the smooth approximation of the conditional density with KDE make the methodology more suitable for selecting representative scenarios with the loss occurrence approach. For this approach, the classical loss disaggregation has to rely on the numerical derivative of the empirical CDF (Baker et al., 2021).

Although single representative scenarios are valuable for risk mitigation and communication purposes, they also have several limitations. For example, designing effective risk mitigation strategies, such as resource allocation before the event, using a single representative scenario would result in solutions tailored to the spatial distribution of damage of the specific selected scenario. Thus, better strategies could be defined by considering multiple scenarios, even for the same loss return period.

Possible extensions of the methodology include catalogs with multiple hazards (e.g., seismic scenarios with tsunami), loss calculations considering indirect consequences, and high-dimensional scenarios (e.g., including the damage states of the individual components, either buildings or power network components). For the later, however, the dimensionality of the damage states has to be reduced (Rosero-Velásquez and Straub, 2019).

# 8 Conclusion

We present a methodology and algorithm to determine representative earthquake scenarios from a synthetic earthquake catalog. We applied the methodology to the communes of Valparaíso and Viña del Mar in Chile. Because the identified scenarios should be representative of extreme losses, they differ depending on the exposed assets. In this contribution, we consider the building stock and the electrical supply network. The application shows that the methodology can work and allows the identification of scenarios more systematically than by selection or extrapolation from past events. However, the results for the building portfolio also show that resulting scenarios cannot be considered independently of the resulting ground motions. Therefore, future work should investigate scenarios that also include the ground motions. Because the description of ground motion fields requires a large number of parameters, the existing methodology will need to be extended to be able to cope with such scenarios.

## Appendix A:  Lists of variables and acronyms

**Table A1.** List of acronyms used in this paper

| Acronym | Name |
|---------|------|
| PGA | Peak ground acceleration |
| PDF | Probability density function |
| PSHA | Probabilistic seismic hazard analysis |
| GMM | Ground motion model |
| CDF | Cumulative distribution function |
| AL | Active learning |
| KDE | Kernel density estimation |
| GPR | Gaussian process regression |
| AEI | Augmented expected improvement |
| INE | Chilean National Statistics Institute |
| UNESCO | United Nations Educational, Scientific and Cultural Organization |
| SEN | Chilean National Electric System |
| ONEMI | Chilean National Office of Emergency at the Ministry of Interior |
| USGS | United States Geological Survey |
| IS | Importance sampling |
| ENS | Energy not supplied |
| DCOPF | Direct current optimal power flow |
| RP | Return period |
| USD | United States dollar |
| ODEPLAN | Chilean National Planing Office |
| CSN | National Seismological Center of the University of Chile |

**Table A2.** List of random variables used in this paper, distinguishing the random variable notation from a value and a sample value.

| Random variable | Value | Sample | Name |
|---|---|---|---|
| $\Theta = [\Theta_1, \cdots, \Theta_d]$ | $\theta = [\theta_1, \cdots, \theta_d]$ | $\theta^{(i)} = [\theta_1^{(i)}, \cdots, \theta_d^{(i)}]$, for $i = 1, \cdots, n$ | Source parameters |
| $L$ | $l$ | $l^{(1)}, \cdots, l^{(n)}$ | Loss |

**Table A3.** List of indices used in this paper

| Notation | Range | Name |
|---|---|---|
| $i$ | $1, \cdots, n$ | Scenario index |
| $j$ | $1, \cdots, n_2$ | Secondary scenario index |
| $k$ | $1, \cdots, n_2$ | Auxiliary secondary scenario index |
| $c_{i,j}$ | $1, \cdots, n$ | Index of the $j$-th closest scenario to $\theta^{(i)}$ |
| $i_t, i_t^{exc}$ | $1, \cdots, n$ | Index of the representative scenario $\hat{\theta}_t$ (resp. $\hat{\theta}_t^{exc}$) |
| $t$ | $t_1, \cdots, t_{n_t}$ | Return period |

**Table A4.** List of numerical parameters used in this paper

| Notation | Name |
|---|---|
| $n$ | Number of scenarios |
| $n_2$ | Number of conditional loss evaluations |
| $n_3$ | Number of AL steps |
| $n_b$ | Number of bootstrap samples |
| $n_s$ | Number of new scenarios to evaluate in an AL step |
| $n_l$ | Number of additional loss evaluations per scenario in an AL step |
| $n_d$ | Number of consecutive converging active learning steps |
| $n_t$ | Number of return periods |
| $n_{train}$ | Training set size for GPR |
| $d$ | Number of source parameters |
| $n^{(i)}$, for $i = 1, \cdots, n$ | Number of conditional loss evaluations computed at $\theta^{(i)}$ |
| $\omega_1, \cdots, \omega_n$ | Scenario weight for Monte Carlo estimations |
| $w_{i,j}$ for $i = 1, \cdots, n, j = 1, \cdots, n_2$ | Scenario weight for KDE |
| $d_{i,j}$ for $i = 1, \cdots, n, j = 1, \cdots, n_2$ | Mahalanobis distance between scenarios $\theta^{(i)}$ and $\theta^{(c_{ij})}$ |
| $c_{neigh}^{(i)}$, for $i = 1, \cdots, n$ | Quality factor of KDE at $\theta^{(i)}$ |
| $\gamma$ | KDE bandwidth |
| $r$ | AL convergence parameter |
| $\epsilon$ | AL convergence threshold |

**Table A5.** List of variables and functions used in this paper. Variables with hat (e.g., $\hat{l}_t$) represent approximations of their respective variables without hat (e.g., $l_t$). Variables with superscript $exc$ refer to the exceedance approach, and the corresponding variables without such this superscript correspond to the occurrence approach.

| Notation | Name |
| --- | --- |
| $t, t_1, \cdots, t_{n_t}$ | Return periods |
| $\lambda_H$ | Earthquake occurrence rate |
| $\lambda_L(l), \hat{\lambda}_L(l)$ | Loss exceedance rate |
| $M_w$ | Earthquake moment magnitude |
| $R$ | Hypocentral distance |
| $\theta_t, \theta_t^{exc}, \hat{\theta}_t, \hat{\theta}_t^{exc}$ | Representative scenario for $t$-year return period |
| $f_A(a), \hat{f}_A(a), f_{A|B}(a|b), \hat{f}_{A|B}(a|b)$ | (Conditional) PDF of $A$ (given $B = b$) |
| $F_A(a), \hat{F}_A(a), F_{A|B}(a|b), \hat{F}_{A|B}(a|b)$ | (Conditional) CDF of $A$ (given $B = b$) |
| $z_t(\theta)$ | $z_t(\theta) = f_{L|\Theta}(l_t|\theta) f_\Theta(\theta)$ |
| $z_t^{exc}(\theta)$ | $z_t^{exc}(\theta) = \left(1 - F_{L|\Theta}(l_t|\theta)\right) f_\Theta(\theta)$ |
| $z_t^{(i)}$ | Approximation of $z_t(\theta^{(i)}) \approx z_t^{(i)} = \hat{f}_{L|\Theta}(\hat{l}_t|\theta) f_\Theta(\theta)$ |
| $z_t^{exc(i)}$ | Approximation of $z_t^{exc}(\theta^{(i)}) \approx z_t^{exc(i)} = \left(1 - \hat{F}_{L|\Theta}(\hat{l}_t|\theta)\right) f_\Theta(\theta)$ |
| $Z_t^{(i)}, Z_t^{exc(i)}$ | Random variable of $z_t^{(i)}$ (resp. $z_t^{exc(i)}$) |
| $\mu_{Z_t}^{(i)}, \mu_{Z_t}^{exc(i)}, \sigma_{Z_t}^{(i)}, \sigma_{Z_t^{exc}}^{(i)}$ | Mean and standard deviation of $Z_t^{(i)}$ (resp. $Z_t^{exc(i)}$) |
| $\theta^*, \theta^{*,exc}$ | Scenario with optimal value of $z_t^{(i)}$ (resp. $z_t^{exc(i)}$) in an AL step |
| $z_t^*, z_t^{*,exc}$ | Value of $z_t^{(i)}$ (resp. $z_t^{exc(i)}$) at $\theta^*$ (resp. $\theta^{*,exc}$) |
| $AEI, AEI(\theta^{(i)})$ | Augmented expected improvement (evaluated at $\theta^{(i)}$) |

**Table A6.** List of additional variables used in this paper for the examples and applications in Sections 4 and 5.

| Notation | Name |
| --- | --- |
| $M_w, m_w$ | Earhquake moment magnitude (as random variable and value) |
| $(X, Y), (x, y)$ | Longitude, latitude (as random variable and value) |
| $H$ | Depth |
| $\alpha, \beta$ | Beta distribution parameters |
| $\mu_{\ln L}, \mu_{\ln L}(m_w), \sigma_{\ln L}$ | (Conditional) mean and standard deviation of the log-losses (given $M_w = m_w$) |
| $\mu_\Theta, \Sigma_\Theta$ | Mean vector and covariance matrix of $\Theta$ |
| $\sigma$ | Noise standard deviation |

*Code and data availability.* The code for scenario selection is available from the authors upon reasonable request. The USGS ComCat Catalog is accessible on the USGS website (https://earthquake.usgs.gov/data/comcat/, last visited: Feb. 2024). The historic earthquakes database of the National Seismological Center of Chile is on the CSN website (https://www.sismologia.cl/informacion/grandes-terremotos. html, last accessed: Feb. 2024). The 2017 Chilean Census, organized by the Instituto Nacional de Estadísticas (INE) is available online

(http://www.censo2017.cl/, last visited: Feb. 2024). The technical information about the SEN, the Chilean power transmission network, is accessible through the website Infotecnica of the National Electrical Coordinator (https://infotecnica.coordinador.cl/, last visited: Feb. 2024).

*Author contributions.* This paper was conceptualized by HRV and DS. The methodology was developed by HRV and DS. The investigation for the hazard modeling (i.e., source model, seismic catalog) was conducted by AP, JCG, JCL, HRV and DS. The investigation for the power supply application (i.e., network model and numerical simulations) was conducted by MM, EF, JCL and HRV. The investigation for the residential building stock (i.e., exposure and vulnerability model and numerical simulations) was conducted by JCG and HRV. The visualization was done by HRV, and the interpretation by HRV and DS. The original draft preparation was done by HRV, MM and DS. Review and editing were done by HRV, AP, JCG, JCL, MM, EF and DS. Funding acquisition for this work was done by DS and JCL.

*Competing interests.* The contact author has declared that none of the authors has any competing interests.

*Acknowledgements.* This work has been sponsored by the research and development project RIESGOS 2.0 (Grant No. 03G0905A-H), funded by the German Federal Ministry of Education and Research (BMBF) as part of the funding programme "BMBF CLIENT II - International Partnerships for Sustainable Innovations", and by the Chilean government through the Research Center for Integrated Disaster Risk Management (CIGIDEN), ANID/FONDAP/1523A0009, the research project Multiscale earthquake risk mitigation of healthcare networks using seismic isolation, ANID/FONDECYT/1220292. We also thank Prof. Fabrice Cotton (GFZ) for his support during the elaboration of this study.

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
