# Peer review of "Risk-informed representative earthquake scenarios for Valparaíso and Viña del Mar, Chile"

_Natural Hazards and Earth System Sciences, 2023_

## Author Comment (AC1)

**Reply to reviewer #1 of "Risk-informed representative earthquake scenarios for Valparaíso and Viña del Mar, Chile"**

Reviewer comments are repeated here in black, our response is in blue font. Text from the paper is given in *italics.*

RC1: 'Review of nhess-2023-186', Anonymous Referee #1, 19 Nov 2023

The manuscript aims to select representative earthquake scenarios for the communication of seismic risk and for the planning of risk mitigation actions. The authors define these scenarios as the scenarios which are most likely to cause a certain regional loss value. For this purpose, they apply a methodology that was proposed by two of the authors in a separate manuscript. This methodology is claimed to require less simulation runs than a conventional loss disaggregation, which commonly relies on a large stochastic event catalog.

The topic is interesting and within the scope of NHESS. The manuscript is well written and shows the expertise of the authors. It is also clear and generally well-structured. Yet, some parts of the manuscript would benefit from additional explanations and discussions to improve its clarity and its contribution to the field. Please see some suggestions below.

We thank the reviewer for their positive feedback and useful comments that help us in clarifying our contributions and in improving the manuscript.

**Major comments**

1. The y-year loss refers to the loss value that is (on average) exceeded every y years. Importantly, it does not refer to the loss value, which (on average) occurs every y years. The identified scenario, however, relates to the most likely scenario that causes this loss value (and not an exceedance of this loss value). Therefore, the link between the return period y and the identified scenario is not straightforward. I am sure that the authors are well aware of this important difference, but the readers (and more importantly, potential users of the identified scenarios) would certainly benefit from additional explanations on this aspect.

   The loss value $l_t$ is the t-year loss, i.e., the loss that on average is exceeded every t years, as the reviewer states. It is selected based on the loss exceedance function. We define a representative earthquake scenario as the mode of the conditional density given this loss, i.e., the occurrence of the $l_t$ loss. However, one can also define it as the mode of the conditional density given the exceedance of that loss.

   A discussion about the differences between the exceedance and occurrence approaches is presented in Fox et al (2016) for hazard disaggregation. Works on loss disaggregation (Goda et al, 2009; Jayaram & Baker, 2009) use the exceedance

approach. To the knowledge of the authors, there are no references discussing the differences in the exceedance and occurrence approach for loss disaggregation.

Realizing that gap, we modified the paper sections to provide better insight into this difference. For example, in the introduction part:

*The above concepts were extended to loss disaggregation to find earthquake scenarios in terms of magnitude and hypocentral distance that exceed a loss threshold for building stocks (Goda and Hong, 2009) or infrastructure (Jayaram and Baker, 2009b) (...). The definition of Rosero-Velásquez and Straub (2022) differs from the loss disaggregation presented by Goda and Hong (2009) and Jayaram and Baker (2009b) because the latter define the representative scenario as the most likely one to exceed the t-year loss. In this contribution, we compare the two definitions and argue that a definition in terms of the occurrence of the t-year loss is more consistent in most cases.*

In Section 2, we introduce the definition in terms of exceedance:

*An alternative definition can be formulated in terms of loss exceedance instead of loss occurrence:*

$$\theta_t^{exc} = \max_{\theta} f_{\Theta|L}(\theta|L \geq l_t) \qquad (4)$$

*In such case, Eq. (4) defines the scenario that is most likely to exceed $l_t$. This is the definition corresponding to the classical loss disaggregation proposed by Goda and Hong (2009) and Jayaram and Baker (2009b). We note that with this definition, in general, the scenario representative of a t-year loss will have a loss return period higher than t. Hence, we find its interpretation more difficult, and prefer the definition in Eq. (3). A similar observation was made by Fox et al. (2016) for the case of hazard disaggregation. Nevertheless, we propose algorithms to evaluate the representative scenarios according to the two definitions and compare the resulting scenarios.*

2. To resolve the above-mentioned issue, the scenario could be identified as the one that contributes most to the losses larger than the y-year loss, ly. In other words, one would aim to find the mode of p(theta | L > ly) rather than the mode of p(theta | L = ly). For hazard disaggregation, for example, Fox et al. (2016) note that such an exceedance-based approach is preferable if one aims to establish a direct link to a ground motion with a specified return period. The results, shown in Figure 7, suggest that the authors performed loss simulations for the entire stochastic event catalog. With these simulations, it should be straightforward to perform such an exceedance-based disaggregation and to compare the resulting scenarios with the already identified ones. Such a comparison would be very valuable and, given that it does require very little additional effort, I recommend that the authors

include a discussion on this aspect in the main body of the paper. The detailed results of this comparison could be shown in an appendix.

This is a recommendation also made by Fox (2023). More specifically, we believe the reviewer might refer to the following statement made in Fox et al (2016): "The exceedance approach is consistent with conventional response history analyses, which are used to determine the expected response for ground motions defined by a certain return period for exceedance". However, Fox et al (2016) also state that "perhaps because of its ready availability, this form of disaggregation is often used directly as an aid in ground-motion selection in PBEE. However, this is not consistent with the typical seismic analysis that follows, which is used to determine the response of a structure at a given intensity (i.e. for $Sa = x_i$ and not $Sa > x_i$, as required in the PEER PBEE framework)". It should be noted that Fox (2023) did not find significant differences between the exceedance and occurrence approaches in the disaggregation results applied to a real case in New Zealand. However, it could be instructive to compare both approaches in our case. Whereas he limits the analysis to hazard disaggregation, we here follow this recommendation for loss disaggregation.

Following your comment, we computed the representative earthquake scenarios with the exceedance approach for the toy example and – as expected – found that they are slightly more extreme than the ones computed with the occurrence approach:

*(a) Loss occurrence approach*                    *(b) Loss exceedance approach*

[Figure]

[Figure]

*Figure 4. Numerical approximation of the representative earthquake scenarios. On panel (a) the representative scenarios computed with the loss occurrence approach, $\hat{\theta}_t$, and on panel (b) the ones computed with the loss exceedance approach, $\hat{\theta}^{exc}_t$. The representative earthquake scenarios correspond to four different return periods t = 50, 100, 500, 1000 years, based on a Monte Carlo sample of scenarios. Each return period is represented by a different color. For each return period, the 20 approximations $\hat{\theta}_t$ (resp. $\hat{\theta}^{exc}_t$), corresponding to 20 experiments, are the colored empty circles, and the corresponding exact solutions are depicted by filled circles. The grey points are the*

*scenarios of the catalog, and the dashed contours represent the PDF of the source parameters.*

Our findings go in line with those of Fox et al (2016) for hazard disaggregation. We have extended the discussion section of the paper to make the case why we consider the occurrence approach to be more suitable for scenarios defined in terms of loss return periods:

*We presented the evaluation of representative earthquake scenarios based on the loss occurrence and the loss exceedance approach; the latter coincides with the loss disaggregation method (Goda and Hong, 2009; Jayaram and Baker, 2009b). In the illustrative example of Section 4.2, we compare the results of the two approaches. For the case of hazard disaggregation, it has been proposed in the literature that the results of both approaches should be reported (Fox, 2023). However, we decided against reporting the scenarios of the exceedance approach for the Valparaíso and Viña del Mar communes, to avoid confusion. We find the loss occurrence approach to have a more intuitive interpretation. Scenarios identified with this approach correspond to a loss that is the t-year loss, which can be reported jointly with the scenarios. They are the most likely scenarios leading to this value (which on average is exceeded once in t years). By contrast, we find it difficult to communicate the meaning of the scenarios with the loss exceedance approach, and we believe it will be mostly misunderstood. Scenarios obtained with the loss occurrence approach can be described as "representative of a loss that is exceeded on average once in t years". For the loss exceedance approach, one would need to describe scenarios as "representative of the losses that would occur when conditioning on a loss at least as large as the one that would be exceeded once in t years", which seems too convoluted to communicate effectively. Nor is it easy to conceive of a risk management activity for which such a definition would be more appropriate.*

3. I agree that scenario-based analyses are valuable for many risk mitigation actions, as well as for risk communication purposes. Yet, the manuscript would benefit from some comments on potential pitfalls related to the use of a single representative scenario. For example, emergency managers may use the estimated spatial distribution of damage and losses from a representative scenario to optimize the placement of machinery and personnel before an event. What if another event (with a different distribution of damage and losses) is almost as likely to cause the considered return period loss. Is the proposed methodology capable of identifying such alternative scenarios?

We added a paragraph in the discussion section to address this point:

*Although single representative scenarios are valuable for risk mitigation and communication purposes, they also have several limitations. For example, designing effective risk mitigation strategies, such as resource allocation before the event, using a single representative scenario would result in solutions tailored to the spatial distribution of damage of the specific selected scenario. Thus, better strategies could be defined by considering multiple scenarios, even for the same loss return period.*

We had also written in the discussion (L408-409) that *for practical risk management tasks it is recommended to use the historic events jointly with the identified scenarios, in particular for the residential building stock case.* That is, the proposed methodology can find alternative scenarios to historical ones, and should complement (not replace) reference scenarios utilized for risk management and communication tasks.

4. A large part of the manuscript focuses on the description of the scenario selection algorithm, which is quite different to the conventional loss disaggregation approach. Yet, the conclusion section contains very little – if any – information on the advantages and limitations of the proposed method (in comparison with the conventional one). I recommend that the authors try to improve the clarity of their conclusions. The summary in the discussion section is much appreciated.

We extended the discussion section to address this point:

*To evaluate the representative scenarios, we adapted the methodology of Rosero-Velásquez and Straub (2022). The methodology leads to lower computational cost in terms of loss evaluations compared to the classical loss disaggregation. By incorporating active learning, the methodology concentrates the conditional loss evaluations around the scenarios that most likely produce the t-year loss value $l_t$. This concentration of samples around the solution and the smooth approximation of the conditional density with KDE make the methodology more suitable for selecting representative scenarios with a loss occurrence approach. For this approach, the classical loss disaggregation has to rely on the numerical derivative of the empirical CDF (Baker et al., 2021).*

We also added a dedicated section called "Computational costs" in the results:

*In terms of loss evaluations, we required one evaluation per scenario in the catalog, for constructing the loss exceedance function with event-based earthquake risk assessment. That corresponds to $2 \times 10^4$ loss evaluations. In addition, during the AL stage, around 10 iterations were necessary to achieve the convergence criterion of Eq. (25), each of them consisting of 160 new loss evaluations ($n_s$ = 2 scenarios evaluated $n_l$ = 20 times, for each of the $n_t$ = 4 return periods). Therefore, 1600 loss evaluations are needed to find the representative earthquake scenarios for 4 different return periods.*

*For comparison, Goda and Hong (2009) report that they use a total of $5 \times 10^6$ loss evaluations for the "classical" loss disaggregation. Furthermore, they only evaluate the scenarios only with the loss exceedance approach. Extending the loss disaggregation approach to the loss occurrence approach will likely require additional evaluations. Additionally, the computation cost of the loss disaggregation approach scales exponentially with the number of parameters describing seismic scenarios. Hence the approach will not be applicable to problems in which earthquake scenarios are described by more than 3 or 4 parameters.*

**Minor comments**

1. Over the past years, the term Ground Motion Model (GMM) seems to be more commonly used than the term Ground Motion Prediction Equation (GMPE). Mainly, because most of the modern empirical GMMs do no longer consist of a single equation. See also the notes of David Boore for some terminological discussion (Boore, 2020), and feel free to adapt it.

   We replaced GMPE with GMM.

2. Lines 94-97: Do the authors have any quantitative comparison (in terms of computation time) between the two loss disaggregation methodologies? This would be interesting and certainly help to highlight the advantages of the proposed method.

   Please refer to the reply to major comment Nr. 4.

**References**

Boore, D. M. (2020): Thoughts on the acronyms "GMPE", "GMPM", and "GMM", available at https://daveboore.com/daves_notes.html

Fox, M. J., Stafford, P. J., and Sullivan, T. J. (2016): Seismic hazard disaggregation in performance-based earthquake engineering: occurrence or exceedance?, Earthquake Engineering & Structural Dynamics

Fox, M. (2023). Considerations on seismic hazard disaggregation in terms of occurrence or exceedance in New Zealand. Bulletin of the New Zealand Society for Earthquake Engineering, 56(1), 1–10

---

## Author Comment (AC2)

**Reply to reviewer #2 of "Risk-informed representative earthquake scenarios for Valparaíso and Viña del Mar, Chile"**

Reviewer comments are repeated here in black, our response is in blue font. Text from the paper is given in *italics*.

RC2: 'Comment on nhess-2023-186', Anonymous Referee #2, 20 Nov 2023

The manuscript with title "Risk-informed representative earthquake scenarios for Valparíso and Viña del Mar, Chile" has as objective setting a criterion for selecting earthquake scenarios to carry out specific studies and activities around disaster risk management and reduction. Overall, the paper is well written and structured, although the manuscript could benefit from adding additional explanations and definitions of multiple concepts and input data which are required for its application. My recommendation is to carry out a major revision before it can be accepted.

We thank you for the positive feedback and your helpful comments. We have considered them in preparing the most recent version of the manuscript. We reply to your comments in the following paragraphs.

Below, I provide general and specific comments that authors may find useful.

**General comments**

1. My main concern is that the manuscript does not include in any section a clear explanation of why this proposed approach is better than the "classic" loss disaggregation obtained from an event loss table and used, for over 10 years, in the catastrophe risk modelling field.

   This point was also brought up by another reviewer, and we now contrast and compare our approach to the "classic" loss disaggregation, as proposed by Goda and Hong (2009) and Jayaram and Baker (2009b), see also our response to major comments 2 and 4 of reviewer 1. We now evaluate scenarios with both approaches. As we discuss, however, we feel that our definition has a more straightforward interpretation.

   Among other texts, we added the following to address this point.

   In the introduction:

   *The above concepts were extended to loss disaggregation to find earthquake scenarios in terms of magnitude and hypocentral distance that exceed a loss threshold for building stocks (Goda and Hong, 2009) or infrastructure (Jayaram and*

*Baker, 2009b) (...). The definition of Rosero-Velásquez and Straub (2022) differs from the loss disaggregation presented by Goda and Hong (2009) and Jayaram and Baker (2009b) because the latter define the representative scenario as the most likely one to exceed the t-year loss. In this contribution, we compare the two definitions and argue that a definition in terms of the occurrence of the t-year loss is more consistent in most cases.*

In Section 2:

*An alternative definition can be formulated in terms of loss exceedance instead of loss occurrence:*

$$\theta_t^{exc} = \max_{\theta} f_{\Theta|L}(\theta | L \geq l_t) \quad (4)$$

*In such case, Eq. (4) defines the scenario that is most likely to exceed $l_t$. This is the definition corresponding to the classical loss disaggregation proposed by Goda and Hong (2009) and Jayaram and Baker (2009b). We note that with this definition, in general, the scenario representative of a t-year loss will have a return period higher than t. Hence, we find its interpretation more difficult, and prefer the definition in Eq. (3). A similar observation was made by Fox et al. (2016) for the case of hazard disaggregation. Nevertheless, we propose algorithms to evaluate the representative scenarios according to the two definitions and compare the resulting scenarios.*

In the discussion section:

*We presented the evaluation of representative earthquake scenarios based on the loss occurrence and the loss exceedance approach; the latter coincides with the loss disaggregation method (Goda and Hong, 2009; Jayaram and Baker, 2009b). In the illustrative example of Section 4.2, we compare the results of the two approaches. For the case of hazard disaggregation, it has been proposed in the literature that the results of both approaches should be reported (Fox, 2023). However, we decided against reporting the scenarios of the exceedance approach for the Valparaíso and Viña del Mar communes, to avoid confusion. We find the loss occurrence approach to have a more intuitive interpretation. Scenarios identified with this approach correspond to a loss that is the t-year loss, which can be reported jointly with the scenarios. They are the most likely scenarios leading to this value (which on average is exceeded once in t years). By contrast, we find it difficult to communicate the meaning of the scenarios with the loss exceedance approach, and we believe it will be mostly misunderstood. Scenarios obtained with the loss occurrence approach can be described as "representative of a loss that is exceeded on average once in t years". For the loss exceedance approach, one would need to describe scenarios as "representative of the losses that would occur when conditioning on a loss at least as large as the one that would be exceeded once in t years", which seems too convoluted to communicate effectively. Nor is it easy to conceive of a risk management activity for which such a definition would be more appropriate.*

*To evaluate the representative scenarios, we adapted the methodology of Rosero-Velásquez and Straub (2022). The methodology leads to lower computational cost in terms of loss evaluations compared to the classical loss disaggregation. By incorporating active learning, the methodology concentrates the conditional loss evaluations around the scenarios that most likely produce the t-year loss value $l_t$. This concentration of samples around the solution and the smooth approximation of the conditional density with KDE make the methodology more suitable for selecting representative scenarios with a loss occurrence approach. For this approach, the classical loss disaggregation has to rely on the numerical derivative of the empirical CDF (Baker et al., 2021).*

2. I suggest authors to review previous works on different topics covered by this manuscript and include several references that in my opinion are missing. I will provide examples of this in the specific comments section.

   We address the suggestions in the specific comments section. We also added some additional literature beyond those pointed out by the reviewer.

3. For the application of this methodology, an event-based earthquake risk assessment must be carried out. However, this is never mentioned or explained in detail and the equations that show how the loss computations are performed are not explicit enough for this.

   We added a sentence in Section 2 to link the calculation of the loss-exceedance function and event-based earthquake risk assessment more explicitly:

   *Because of the randomness and uncertainty in the earthquake scenario, GMM, vulnerabilities, and exposure, L is a random variable whose cumulative distribution function (CDF) $F_L(l)$ can be obtained by performing an event-based earthquake risk assessment for spatially distributed systems with the synthetic earthquake catalog,...*

   Additionally, we now introduce event-based earthquake risk assessment in the same section, as we detail in our response to the specific comment Nr. 11.

4. It is not clear why in the case study, two (very) different synthetic earthquake catalogs are used. What is the benefit of doing so? Are the results at any stage combined?

   We compare the loss exceedance function associated with the building stock computed with two synthetic earthquake catalogs and choose the catalog that best represents historical evidence of losses. As a result, we observed that the nationwide catalog represents better the historical evidence of losses.

   In addition, the SARA catalog only considers events near Valparaíso and Viña del Mar. Although it is sufficient for the building stock, it is not for the power network, which covers the entire country. Furthermore, the magnitude in the SARA catalog was sampled with standard Monte Carlo. In contrast, in the

nationwide catalog, the magnitude was sampled with Importance Sampling, thus it sampled extreme events more efficiently.

However, since we only compared the catalogs with the loss exceedance functions associated with the building stock, and since the paper now has more focus on comparing and discussing two approaches of loss disaggregation (see also the reply to general comment 1 and the replies to major comments 1 and 2 of Reviewer 1), we decided to report only results with the nationwide catalog, which now is simply referred to as the synthetic earthquake catalog.

5. There are different statements made by the authors that are not accompanied by evidence or references. I will provide examples of this in the specific comments section.

   We answer to the specific comments below. We also read the paper again and added additional references and evidence in selected places.

6. Some of the conclusions of the paper are contradictory, between the two case studies.

   The discussion and conclusion of the paper were significantly edited, so we hope that the contradictions identified by the reviewer were removed in the process.

7. Authors in my understanding are referring interchangeably to synthetic earthquake catalogs and stochastic event sets, whereas these are two very different representations. I suggest that the difference between them is explicitly mentioned, and also which representation is the one used in the proposed methodology.

   We now specify throughout the manuscript that we work with a synthetic earthquake catalog, and we keep that term since it emphasizes the fact that the events are generated through a numerical model. However, we note that following Baker et al (2021, p. 280), as well as the cited references there, the terms "synthetic earthquake catalog" and "stochastic event set" may also be used interchangeably.

8. Case studies in 4.1 and 4.2 show an example for a single building. However, the EQ loss assessment explained with the equations in the paper is not a good approach for this type of assessment and is usually preferred only for portfolio assessments.

   The case studies are highly idealized to demonstrate the principle of the methodology. The systems there could represent a single building, but also a system or portfolio (at an abstract level).

*We reworded the descriptions of both case studies in such a way that they represent hypothetical building portfolios instead of single buildings.*

9. A discussion about how the methodology performs in a case with multiple buildings and how the treatment of the spatial correlation may introduce changes with respect to the results of the two case studies presented.

   *The application of the methodology to the building stock and power network in the communes of Valparaíso and Viña del Mar in Chile, presented in Sections 5 and 6, illustrates the application of the methodology in the case of multiple assets (including buildings). The following discussion in Section 7 analyzes the results considering the correlation effects in the building stock (concentrated assets) and power network (sparse assets):*

   *(...) According to the employed model, extreme losses are more likely to occur by a combination of a less strong earthquake with larger-than-average ground motions (i.e., a large value of the inter-event term in the GMM). This effect occurs for the residential building stock, due to its spatial concentration, and not (or to a much smaller extent) for the power supply network, which is spatially distributed.*

   *Regarding the treatment of spatial correlation, a discussion about the influence of the spatial correlation in loss disaggregation in a building stock can be found in Gómez-Zapata et al (2022a), which we cite in the manuscript. In addition, we added a paragraph in Section 5.2.1:*

   *Different authors have also analyzed the effects of the spatial correlation in the losses (Goda and Hong, 2009; Jayaram and Baker, 2009a, b; Baker et al., 2021), and they conclude that it should be included in spatially distributed systems, otherwise, the loss exceedance function underestimates extreme events.*

10. Some decisions/assumptions made by the authors are not very clear. As for instance, why if the two case studies are located within the same area/country, different GMMs are used for each of them?

    *The models for the power network and building stock were developed independently by CIGIDEN in Chile (power network) and GFZ (building stock). One of the differences in the models is the choice of the GMM. However, as explained in L310-L315, their functional form is similar, and therefore, their predictions do not differ significantly, as observed in previous studies (e.g., Hussain et al., 2020; Gómez-Zapata et al., 2022a). In particular, Hussain et al. (2020) found negligible differences in direct loss estimates for the residential building stock of Santiago de Chile after using these two GMMs to simulate the associated ground motion from subduction earthquake scenarios.*

**Specific comments**

1. At the abstract, I suggest changing "risk management tasks" for "risk management activities".

   *Amended.*

2. At the abstract (and the introduction), it must be explained why the mentioned activities make use of scenarios of earthquake events.

   *We decided to keep the abstract short but added an additional explanation to the introduction part: Scenario-based analysis enables the modeling and simulation of the complex processes and interactions during and after earthquake events, with a level of detailing that is not possible in a complete probabilistic hazard and risk analysis.*

3. In the abstract it says that earthquake scenarios are defined in terms of the loss exceedance. Is this referring to rates? Probabilities? If the latter, in which timeframes?

   *We modified the abstract to clarify that it refers to annual loss exceedance rates.*

4. The introduction mentions that earthquake scenarios are the starting point for detailed risk assessments. However, this is not true nowadays and even more, today it is more common to carry out a fully probabilistic and event-based EQ risk assessment, and from the results (e.g., ELT), choose events to carry out scenario analyses.

   *We modified the original sentence to make clear that "detailed risk assessment" refers to scenario analyses.*

5. The classic PSHA formulation by Esteva (missing reference) and Cornell, did not aim to generate synthetic earthquake catalogs or stochastic event sets (note that these two are not the same). This statement at the introduction must be revised and adjusted.

   *We revised the statement and the context in which it was written. The paragraph aims to review studies where one of the products is a set of selected earthquake scenarios, including cases when the selection is within a synthetic earthquake catalog. The classic PSHA formulation is still far from scenario selection; some intermediate steps, such as Monte Carlo PSHA and PSHA products (Baker et al, 2021), were missing in our text and this is now clarified.*

   *Therefore, we moved the sentence to Section 2 to a new paragraph:*

   *Synthetic earthquake catalogs have been used in event-based probabilistic seismic hazard analysis (PSHA) and earthquake risk assessment (e.g., Salgado-Gálvez et al.,*

*2018; Ferrario et al., 2022; Allen et al., 2022). PSHA aims to obtain the occurrence rate and distribution of ground motions, taking into account all possible earthquake scenarios (Cornell, 1968; Esteva, 1970). Event-based PSHA utilizes Monte Carlo simulation for sampling earthquake scenarios. Similarly, event-based earthquake risk assessment on spatially distributed systems utilizes synthetic earthquake scenarios for computing the losses, considering the spatial correlation in the ground motion and the vulnerability of the exposed assets (Baker et al., 2021).*

6. L21: it says that the classic hazard disaggregation does not consider the losses of the affected systems. This is evident and correct since as authors mention, it has to do only with the hazard component. I suggest removing that sentence.

   Albeit evident to the reviewer, the literature and discussion around loss disaggregation, compared to hazard disaggregation, is more limited. Therefore, we prefer to keep that sentence.

7. L25: what is the accumulated loss? Spatially accumulated? Temporal accumulation?

   It is only spatially accumulated loss. We added a clarification in the text.

8. L28: please clarify if the return period mentioned is that one for the loss, and if so, it is worth highlighting that it is usually very different than the one of the event.

   We added a clarification that the return period is the one for the loss:

   *Rosero-Velásquez and Straub (2022) proposed a definition of a representative hazard scenario associated with a loss return period t, e.g., the 100-year loss, which in general does not correspond to the magnitude or intensity measure of the same return period.*

   We decided to keep the distinction from the magnitude and the intensity measure to avoid confusion.

9. L47: I suggest adding "network" after power supply.

   We added it to the text.

10. Authors refer along the text to seismic catalogs. A definition and comprehensive explanation of what these are, what they include, etc. is needed. On L48 for instance, it is not clear if authors are referring to historical catalogs, synthetic catalogs, or both. Only in L106 it is mentioned a "stochastic seismic catalog" which in the cat-risk modelling jargon is not common.

    We adjusted the explanation to clarify that in the paper we work with a synthetic earthquake catalog. Since we define them as a set of earthquake scenarios

(L65), and we explain what we mean by earthquake scenario (L61), we already explain what they include (i.e., source parameters such as hypocentral location and magnitude)

11. Section 2 required adding a better description of event-based PSHA (plus the corresponding appropriate references). Also, I think that in this section is where the explanation between stochastic event-sets and synthetic catalogs must be included.

    Please refer to the reply to specific comment Nr. 5.

12. L67: PGA and Sa are not inputs to assess the vulnerabilities but the losses to the exposed systems.

    We corrected the sentence.

13. Section 2 also requires adding an explanation of event-based EQ risk assessment (including the appropriate references"

    Please refer to the reply to specific comment Nr. 5.

14. The proposed methodology seems to work well in cases where only one source is controlling the EQ hazard and risk. Some discussion about its applicability in other (more common) contexts where multiple sources contribute to the overall EQ hazard and risk levels is required.

    The methodology also works with more than one seismic source. We clarified in Section 2 that the PDF of the source parameters *θ* "*is obtained from one or more seismic source models*". In addition, the occurrence model that generated the synthetic earthquake catalog utilized for the study area in Chile divides the subduction area into 7 seismic zones, i.e. 7 seismic sources. Therefore, the methodology has been tested in the manuscript with an application that considers multiple sources.

15. The explanation of Eq. 4 starting in L93 is only one way of treating the aleatory uncertainty in probabilistic risk assessments. Others (perhaps more efficient and with similar results) exist and must be mentioned and referenced.

    We modified that explanation in the following way:

    *The objective functions of Eq. (5) and (6) consist of the PDF (...) which can be approximated with conditional samples of losses. One way to account for the aleatory uncertainty in the modeled ground motions is to draw thousands of random samples (Silva, 2016) and propagate them to the loss metrics. However, performing this amount of loss evaluations for an entire seismic catalog (normally containing dozens of thousands of events) is computationally (too) expensive. Alternative ways consider active learning with Gaussian process models (Tomar and*

*Burton, 2021; Rosero-Velásquez and Straub, 2022), or using extreme value theory and the generalized Pareto distribution (Borzoo et al., 2021). Therefore, we propose to first perform only one loss evaluation for each scenario in the catalog (...). This methodology is an adaptation of the one proposed by Rosero-Velásquez and Straub (2022).*

16. A map with the epicenters could accompany Table 1 for a better understanding.

    We added a note in Table 1 indicating that the geographical location of the epicenters is presented in the results, where they are compared with the resulting scenarios of the proposed approach.

17. L279: is that the original or the modified G-R relationship?

    It is the original G-R relationship. However, since the event magnitudes in the synthetic catalog were sampled with Importance Sampling (IS), using a uniform distribution, that relationship is used for computing the IS weights and not for the sampling. We clarified that line in the text.

18. To me, it is not clear what is the purpose of using two (very) different synthetic EQ catalogs and why, if one covers the small (buildings) and larger (power network) areas, the other one is needed. Also, a comparison of the two catalogs (e.g., rates by bins) for the "common area" would be useful if authors decide to keep the two.

    Please refer to our reply to general comment Nr. 4

19. Figure 5: the size of the dots as a function of Mw is not very visible in the maps.

    We are aware of this issue. However, we could not find a better way of displaying the 20 000 scenarios of the catalog on a single map. In any case, one can see in the figure dots of different sizes, even though one cannot determine their exact magnitude. The intention is also to show the effect of the Importance Sampling, and now we highlight this in the text, Section 5.2: *The resulting catalog is depicted in Figure 5, in which one can observe that events of different magnitudes have similar spread within the seven seismic zones.* We now also show in the figure the seven seismic zones defined by Poulos et al (2019).

[Figure]

*Figure 5. Synthetic earthquake catalog with 20 000 scenarios (Poulos et al., 2019). The circle size corresponds to the scenario magnitude. The red square contains the study area. Seismic zones 1 to 3 are of subduction interface type, and zones 4 to 7 are of subduction intra-slab type. Basemap from ©OpenStreetMap contributors 2023. Distributed under the Open Data Commons Open Database License (ODbL) v1.0.*

20. L352: authors state that loss estimations are "similar" from 10^8 onwards, but the results shown in Figure 7 show a very different thing. The EP curves even overlap. Again, in this point is not clear what is the purpose and benefit of using two synthetic EQ catalogs.

    Looking at Figure 7, the 100-year loss is around 3x10^8 according to the curves obtained from both catalogs. The 50-year loss ranges between 1.5 and 2x10^8 USD, the 500-year loss ranges between 2x10^9 and 3x10^9, and the 1000-year loss between 4x10^9 and 5x10^9 USD. Taking the relative error of the log10-losses with respect to the nationwide catalog, the error is below 2% for the four loss quantiles. Therefore, we conclude that the loss estimations between the two exceedance curves are similar (in the sense of their log10s) for the range between the 50-year loss and the 1000-year loss.

    Concerning the issue of using two synthetic earthquake catalogs, please refer to our reply to general comment Nr. 4. In consequence, now Figure 7 only shows the loss exceedance function with the nationwide catalog:

[Figure]

*Figure 7. Loss-exceedance function of the reconstruction costs associated with the residential building stock in Valparaíso y Viña del Mar communes.*

21. L362: a better justification of why the spread is deemed as acceptable is missing.

    We modify the explanation:

    *(...). In all evaluations, we found a spread of the identified representative scenarios, similar to that of Figure 4. This spread is larger for higher return periods, but most of the numerical solutions (11 out of 20 for the 1000-year loss return period and at least 16 out of 20 for the other loss return periods) have epicentral locations within a radius of 50km around the mode, and the coefficient of variation of the magnitude is below 4% for all return periods. (...)*

    And added a similar one for the power network:

    *The spread of the solutions obtained with the 20 runs is larger than the one of the residential building stock in epicentral locations. At least 13 solutions cluster around the sample mode within a radius of 100km, and the coefficient of variation of the magnitude is below 5% for all return periods*

22. The light purple color for RT 100yrs does not contrast well with the grey background in Figures 8, 10 and 11. It could be changed to other tone.

    We modified the figures.

23. The discussion section includes some interesting conclusions and statements that can be better understood if more evidence or explanations are included. For instance, what useful validatíons can be made? (L394). Why the 500 and 1000yr scenarios are larger than the expected? (L397).

We modify the explanation of the referred lines in Section 7:

*(...) The fact that the scenarios identified with the proposed approach differ from the historical events selected in Indirli et al. (2011) should not be surprising, as the latter are in some sense just "random samples" of earthquake events. Nevertheless, the historical events can provide a useful validation of the identified scenarios. In this regard, the scenarios identified as representative of the power supply network appear to be in line with the historic events, as they cover geographic areas of similar sizes, as shown in Fig. 11b. The identified 500 and 1000-year scenarios have larger magnitudes than the historical events, which would also be expected since the historical events all come from a (roughly) 100-year period, see Table 1. By contrast, the representative scenarios identified for the building stock have significantly smaller magnitudes than the historical events. However, they occur much closer to the considered building stock. (...)*

---

## Author Response (AR2)

**Review: Risk-informed representative earthquake scenarios for Valparaíso and Viña del Mar, Chile**

RC1: 'Review of nhess-2023-186', Anonymous Referee #1, 13 Mar 2024

The authors replied to all previous comments of this reviewer and have put substantial effort to address these comments in the improved manuscript. Overall, I am satisfied with the replies and the revised manuscript. I have one minor comment (see below), which addresses a single paragraph of the manuscript. Hence, I am happy to accept the article after these corrections.

Comment: In lines 474-484 of the revised manuscript, the authors make relatively bold statements on why the occurrence-based approach is superior to the exceedance-based approach for loss disaggregation. They claim that the occurrence-based approach is easier to communicate, because it leads to a scenario which is "representative of a loss that is exceeded on average once in t years". While it may be "representative" of such a loss, it is difficult to establish a connection with the return period, because this loss is exceeded, on average, every t years. Thus, the identified scenario is not representative of the losses which may materialize given such an exceedance (this would be better addressed by the exceedance-based approach). Both approaches have their advantages and disadvantages. The presented manuscript does not contain sufficient results and case studies that support the relatively strong statements in lines 474-484. Results for the main case-studies, for example, are only presented for the occurrence-based approach. Thus, I recommend that the authors carefully revise this paragraph. I believe that it warrants further studies to compare both approaches, such that one may give practice-oriented recommendations on which one to use.

We thank you for the positive feedback. We modified the paragraph accordingly:

(...) which seems too convoluted to communicate effectively. *We also have difficulties to conceive of a risk management activity for which such a definition would be more appropriate. However, we acknowledge that this discussion could benefit from additional comparisons of the two approaches in future studies.*

**Review: Risk-informed representative earthquake scenarios for Valparaíso and Viña del Mar, Chile**

RC2: 'Comment on nhess-2023-186', Anonymous Referee #2, 29 Mar 2024

The revised version is acceptable in my opinion and has considered mine, and other Reviewer's comments in an appropriate way.

We thank you for the positive feedback.

---

## Author Response (AR3)

**Review: Risk-informed representative earthquake scenarios for Valparaíso and Viña del Mar, Chile**

'Review of nhess-2023-186', Executive director NHESS, 20 May 2024

Dear Hugo Rosero-Velásquez, Mauricio Monsalve, Juan Camilo Gómez Zapata, Elisa Ferrario, Alan Poulos, Juan Carlos de la Llera, and Daniel Straub:

I am pleased to accept your manuscript "Risk-informed representative earthquake scenarios for Valparaíso and Viña del Mar, Chile" for publication in our journal NHESS subject to some minor revisions with a quick check by the editor of your changes.

We thank you for accepting our manuscript. In the following we reply to the minor revisions

These very minor revisions are as follows:

* Abstract. Please be a tad more rigorous in reporting values for data/methods/results in the abstract. It is currently fairly high level and not really a summary. For example, how many scenarios, what is the size of the synthetic earthquake catalog, down to what magnitude, how many scenarios did you identify? There is a lot more 'summary' that you could put in for your paper to be useful to someone who has 'just' the abstract to read.

We modified the abstract as follows:

(...) With this approach, we identify representative earthquake scenarios for the return periods of 50 yr, 100 yr, 500 yr and 1000 yr in the Chilean communes of Valparaíso and Vi\~na del Mar, based on a synthetic earthquake catalog of 20 000 scenarios on the subduction zone with magnitude $M_w \geq 5.0$. We consider separately the residential building stock and the electrical power network, and identify and compare earthquake scenarios that are representative for these systems. Because the representative earthquake scenarios are defined in terms of the annual loss exceedance rates, they vary in function of the exposed system. The identified representative scenarios for the building stock have epicenters located not further than 30 km from the two communes and magnitudes ranging between 6.0 and 7.0. The epicenter locations of the earthquake scenarios representative for the electrical power network are more spread out, but not further than 100 km away from the two communes, and with magnitudes ranging between 7.0 and 9.0. For risk management activities, we recommend considering the identified scenarios together with historical events.

* I believe this paper will be much more useful to the reader if you have a table of variables and acronyms, and what they mean, that you introduce early on in Section 2. This is because you have a large number of these, and it makes for dense reading.

We included an annex with tables of variables and acronyms.

* Figures 2, 3, . Font size is getting too small to read for some of the text.

We amended the figures accordingly.

* Table 1, text, Figure 5 x- and y-labels, . Be clear when you put degrees in, whether this is E, W, N, S.

We amended the maps and tables accordingly.

* Units. Everywhere you have something like "500 x 500 m" it should read "500 m x 500 m" (you've left off units for first 500. You have m in non-italic and s in italic, in section 5.3. Please throughout your manuscript be consistent (check NHESS guidelines for guidance).

We amended this throughout the text.

* Figure 6a. I found it difficult to differentiate the colours in your legend and then go to the actual figure. For the counts, you have duplicated numbers for your bins (500 appears twice, 1000 appears twice, 1500 appears twice). It should be 0-499, or 501-1000, etc.

We modified the figure considering this.

* Figure 7 and associated text. Wherever you mention USD, you need to be clear what year this is for, as it changes (significantly) with inflation. So you cannot compare USD from one decade with two decades later.

We now specify that the reported monetary values are from 2016, which corresponds to the year when the replacement cost of the considered building types were estimated (Yepes-Estrada et al, 2017). Similarly, we also report a value in USD in 1985 values, corresponding to reported losses due to the 1985 earthquake event.

* Figure 8. If you are going to use RP, then define it in the figure caption (Return Period). I'm not able to easily see the colours of the buildings and differentiate them.

We modified the figure caption accordingly.

* General. (e.g., Figure 10). Black on dark purple is very hard to see.

We amended the figures accordingly.

* The discussion is good, but could be broken out into clear topics, and let the reader know at the beginning what those distinct topics will be.

We now divide the section into two subsections, whose titles tell the reader the two topics (i.e., "On the results for the communes of Valparaíso and Viña del Mar" and "On the definition of representative scenarios and the methodology for identifying them")